# LEARNING RATE GRAFTING:
# TRANSFERABILITY OF OPTIMIZER TUNING

## ABSTRACT

In the empirical science of training large neural networks, the learning rate schedule is a notoriously challenging-to-tune hyperparameter, which can depend on all other properties (architecture, optimizer, batch size, dataset, regularization, ...) of the problem. In this work, we probe the entanglements between the optimizer and the learning rate schedule. We propose the technique of optimizer grafting, which allows for the transfer of the overall implicit step size schedule from a tuned optimizer to a new optimizer, preserving empirical performance. This provides a robust plug-and-play baseline for optimizer comparisons, leading to reductions to the computational cost of optimizer hyperparameter search. Using grafting, we discover a non-adaptive learning rate correction to SGD which allows it to train a BERT model to state-of-the-art performance. Besides providing a resource-saving tool for practitioners, the invariances discovered via grafting shed light on the successes and failure modes of optimizers in deep learning.

## 1 INTRODUCTION

Adaptive gradient methods and learning rate schedules are cornerstones of optimization for deep learning. The pursuit of faster and more robust convergence in training deep neural networks has led to a preponderance of optimizer update rules, annealing heuristics, and hyperparameters, with no clear or principled way to select between them. The difficulty makes tuning a state-of-the-art deep learning pipeline very opaque and expensive.

Consequently, a large body of work aims to understand the structure of these search spaces, analyzing the properties of optimizers and their components in idealized models to predict directional trends, provide rules of thumb, and propose mechanisms for emergent phenomena (like generalization and transfer). Other works study the intrinsic tradeoffs between competing objectives, such as faster convergence vs. generalization. The continued prevalence of black-box hyperparameter search points to a consensus that a quantitatively predictive understanding of training dynamics currently eludes us.

Towards forming more robust beliefs in this space, we provide a recipe that enables a wide class of ablation studies on state-of-the-art models. We consider an unnatural-looking meta-algorithm, which combines the update rules of two different optimizers, forming a single *grafted* update rule. Our findings serve not only as a sanity check on explanatory theory, but also as a way to simplify the optimizer search space in practical settings.

### 1.1 OUR CONTRIBUTIONS

We introduce learning rate grafting, a meta-algorithm which blends the steps of two optimizers by combining the step magnitudes of one ($\mathcal{M}$) with the normalized directions of the other ($\mathcal{D}$). Applied to state-of-the-art deep models, grafting leads to the following:

**Transfer of optimizer performance.** We discover that in many deep learning settings, the performance of an optimizer can be explained by the *implicit step size schedules* it induces. Grafting provides an adaptive way to transfer these schedules between algorithms, consistently closing performance gaps between different optimizers used to train the same model architecture. As a result, grafting provides a strong baseline for learning rate schedule tuning: given a tuned baseline $\mathcal{M}$,

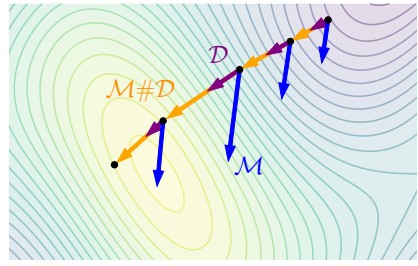

Figure 1: The optimizer grafting operation introduced and studied in this work: meta-optimizer $\mathcal{M}\#\mathcal{D}$ uses the update magnitude of optimizer $\mathcal{M}$, and the direction of $\mathcal{D}$.

grafting can enable a different optimizer $\mathcal{D}$ to match (and sometimes exceed) the performance of $\mathcal{M}$, without tuning the learning rate schedule for $\mathcal{D}$. We demonstrate this phenomenon in multiple settings: across vision and language models, and different batch sizes.

**Schedule discovery.** We show how to use grafting for the exploratory analysis of optimization trajectories and implicit learning rate schedules. Using grafting as a diagnostic tool, we discover a per-layer learning rate correction that allows SGD with momentum (without adaptive preconditioning) to pretrain a BERT model to state-of-the-art accuracy for the first time. This correction is non-adaptive, and does not require the parallel execution of the original algorithm from which schedule is derived. We show analogous results for vision models, finding novel and effective global (as opposed to per-layer) schedules for AdaGrad. We discuss the implications of this for contemporary theory and engineering practices in large-scale settings: the distinctions between widely-used optimizers may boil down to just a few *implicit step size schedules*, not the usual understanding of per-coordinate "preconditioned directions".

## 1.2 RELATED WORK

**Learning rate schedules.** Choosing learning rate schedules is a difficult empirical problem, for which many heuristics exist; the design and search of these schedules can result in dramatic performance differences (Gotmare et al., 2018; Smith, 2018; Smith & Topin, 2019; Loshchilov & Hutter, 2016). In large-scale language modeling, a source of particularly resource-intensive training problems, the predominant empirical practice is to tune a warmup and decay alongside an adaptive optimizer (Popel & Bojar, 2018; You et al., 2019). Beyond classical minimax convergence rates for stochastic optimization, some theory has been established to explain the benefits of certain learning rate schedules. Ge et al. (2019) provide a fine-grained theoretical account for quadratic losses. Agarwal et al. (2021) show that optimized non-adaptive learning rate schedules can induce stable and usable Nesterov-like acceleration. Li & Arora (2019) and Arora et al. (2018) study the interaction of learning rates with batch normalization (Ioffe & Szegedy, 2015), while Li et al. (2019) provide a theoretical mechanism by which initial large learning rates improve generalization. The interaction of learning rates with batch size is explored in (Krizhevsky, 2014; Goyal et al., 2017; Bottou et al., 2018; Shallue et al., 2019). Anil et al. (2020) use the grafting method to stabilize an exotic full-matrix optimizer.

**Adaptive optimizers in deep learning.** Adaptive methods have turned out to be extremely robust in training deep neural networks, receiving tens of thousands of citations for this reason. In particular, Adam has been the *de facto* standard in fields such as NLP (e.g. (Devlin et al., 2018; Yang et al., 2019; Liu et al., 2019)), deep generative modeling (e.g. (Karras et al., 2017; Brock et al., 2018; Kingma & Dhariwal, 2018)), and deep RL (e.g. (Haarnoja et al., 2018)). Adaptive methods have seen adoption in extremely large-scale settings, necessitating modifications to reduce resource consumption (Shazeer & Stern, 2018; Anil et al., 2019; Chen et al., 2019).

An important discussion was sparked by Wilson et al. (2017), who presented empirical and theoretical examples where adaptive methods generalize poorly. Building on this premise, Keskar & Socher (2017) suggest switching from Adam to SGD during training. A variety of theoretical mechanisms have been proposed for the behavior of adaptive methods; for example, learning rate schedule auto-

tuning (Ward et al., 2019), escaping saddle points (Staib et al., 2019), and mitigating long-tailed distributions (Zhang et al., 2019). Reddi et al. (2018) construct, then mitigate, pathological settings where Adam fails to converge. However, in the vast majority of cases, out-of-the-box adaptive methods are perfectly suitable for practitioners. Large-scale empirical studies (Choi et al., 2019; Schmidt et al., 2020; Nado et al., 2021) confirm that hyperparameter search, especially learning rate schedules, are essential to state-of-the-art performance, and undertuned baselines can often lead to faulty optimizer comparisons.

**Layer-wise adaptive methods.** Various algorithms have been proposed recently that are focused on adaptive layer-wise scalings on top of base algorithms. Adaptive layer-wise scalar obtained via weight norms have been derived in LARS/LAMB (You et al., 2017; 2019) and successfully applied to both Momentum and Adam. Several works (Shazeer & Stern, 2018; Anil et al., 2019; Chen et al., 2019) propose compressed preconditioners to reduce the space of per-coordinate adaptive multipliers (as maintained by algorithms such as AdaGrad/Adam) to a factored space. Novograd (Ginsburg et al., 2019) reduces the per-coordinate scalars to per layer scalars via addition. LARS-like scalars on top of normalized SGD with momentum was used to train BERT in Cutkosky & Mehta (2020). To the best of our knowledge, non-adaptive per-layer learning rate corrections, such as those presented in Section 5, have not been explored.

## 2 PRELIMINARIES

### 2.1 OPTIMIZERS AS GENERIC STATE MACHINES

There are various abstractions of optimization in deep learning: for example, we could reason about stochastic first-order oracles $\widehat{\nabla F}(w)$ which are unbiased estimators of some gradient of a population objective $F(w)$. However, this fails to capture various popular deviations from the standard empirical risk minimization model, such as sequential batching in NLP (Merity et al., 2018; 2017) or deep reinforcement learning, where optimization deviates from its theoretical abstractions in many ways (Ilyas et al., 2019; Engstrom et al., 2020).

For this reason, we take the maximally agnostic approach for our notation and definitions in this paper. We choose the *software* abstraction of optimization: we view a gradient-based optimizer as a state machine which takes a sequence of gradients $g_t \in \mathbb{R}^d$ and outputs an update rule for the weights $w_t \in \mathbb{R}^d$. This abstraction is agnostic to properties or theoretical guarantees of the optimizer; in supervised learning, the gradients are usually understood to be averaged over a minibatch, but we do not use this property either.

Thus, an algorithm $\mathcal{A}$ takes the current iterate and gradient $(w_t, g_t)$, and outputs the next iterate $w_{t+1}$; this update can depend on the algorithm's internal state, hyperparameters, side information, or randomness. Then, vanilla SGD with learning rate schedule $\eta_t$ is the algorithm $\mathcal{A}$ specified by

$$\mathcal{A}(w_t, g_t) := w_t - \eta_t g_t. \tag{1}$$

Then, in the usual process of iterative training, the model weights are iteratively updated, as in $w_{t+1} \leftarrow \mathcal{A}(w_t, g_t)$. Even in simple theoretical settings like least-squares regression (Jain et al., 2018; Ge et al., 2019), the choice of the time-varying scalar hyperparameter $\{\eta_t\}$ already presents subtleties and bias-variance tradeoffs.

### 2.2 ADAPTIVE METHODS

We now bring in the second ubiquitous object of study: adaptive gradient methods. Broadly, they refer to the technique of maintaining a sequence of linear transformations $H_t$, dependent on the observed information $(w_t, g_t)$ so far, to modify the (stochastic) gradient descent steps:

$$\mathcal{A}(w_t, g_t) := w_t - \eta_t H_t g_t.$$

Despite their extreme popularity and theoretically principled origins, there is considerable uncertainty regarding the dynamics of adaptive methods. They were first motivated by regret bounds dependent on the geometry of the data (McMahan & Streeter, 2010; Duchi et al., 2011), hence

the terminology "adaptive regularization". The view of loss curvature estimation leads to the term "adaptive preconditioning", where $H_t$ serves as an adaptive scaling factor to make the problem less ill-conditioned (Tieleman & Hinton, 2012; Kingma & Ba, 2014).

Although more exotic adaptive optimizers exist (Martens & Grosse, 2015; Gupta et al., 2018; Agarwal et al., 2019), we focus on diagonal second-moment preconditioning methods, which have yet to be dethroned as the *de facto* standard. This family of optimizers accumulates the squares of the entries of $g_t$, and sets $H_t$ to be the diagonal matrix of entrywise inverse square roots of the accumulators.

**Entanglements between preconditioners and learning rates.** Notice that in Equation 2.2, a scalar multiplier on the preconditioner $H_t$ can be absorbed into the learning rate schedule $\eta_t$. In other words, the dynamics of $H_t$ can induce an *implicit step size schedule* over the course of training, and different choices of optimizers (or configurations of the same optimizer) lead to different implicit schedules. This entanglement motivates the introduction of our grafting meta-algorithm. Specifically, we design an experiment to answer the following question: *does the empirical success of an adaptive optimizer arise from the true preconditioner $H_t$, or can it be equivalently described by a learning rate schedule $\eta_t$?* By combining the step magnitudes of an optimizer $\mathcal{M}$ and step directions of an optimizer $\mathcal{D}$, we can isolate and disentangle adaptive preconditioners $H_t$ from their induced step size schedules.

## 3 THE GRAFTING META-ALGORITHM

We begin by outlining the grafting procedure, which maintains two child optimizers $\mathcal{M}, \mathcal{D}$ and outputs a grafted update step; we call the entire algorithm $\mathcal{M}\#\mathcal{D}$. Before developing the meta-algorithm in full generality, we walk through the execution of the simplest case:

1. At each training iteration, $\mathcal{M}\#\mathcal{D}$ feeds the same input $(w_t, g_t)$ to both children $\mathcal{M}, \mathcal{D}$, which manage their states independently and produce outputs $w_\mathcal{M}, w_\mathcal{D}$.

2. *We do not update $w_{t+1}$ yet.* Instead, we compute $\|w_\mathcal{M} - w_t\|$ and $\|w_\mathcal{D} - w_t\|$, the norms of the steps the child optimizers would have taken.

3. The update step that grafting outputs is

$$\begin{cases} w_t + \frac{\|w_\mathcal{M} - w_t\|}{\|w_\mathcal{D} - w_t\|}(w_\mathcal{D} - w_t) & w_\mathcal{D} \neq w_t \\ w_t & \text{otherwise} \end{cases},$$

which combines $\mathcal{M}$'s update magnitude with $\mathcal{D}$'s update direction.

We remark on a few elementary properties of this binary operation on optimizers:

- Grafting is *idempotent*. If $\mathcal{A}, x_1, g_t$ are deterministic, then $\mathcal{A}\#\mathcal{A} \equiv \mathcal{A}$ (meaning that these two optimizers have the same trajectory under identical sequences of inputs $(w_t, g_t)$).

- If $\mathcal{M}, \mathcal{D}$ differ only by a learning rate schedule (i.e. $w_\mathcal{M} - w_t$ is always a positive scalar multiple of $w_\mathcal{D} - w_t$, under identical inputs $(w_t, g_t)$), then $\mathcal{M}\#\mathcal{D} \equiv \mathcal{M}$.

- Disregarding the special case where $\mathcal{A}$ outputs a zero step, we have

$$(\mathcal{M}\#\mathcal{A})\#\mathcal{D} \equiv \mathcal{M}\#(\mathcal{A}\#\mathcal{D}) \equiv \mathcal{M}\#\mathcal{D}.$$

- Grafting is *not commutative*. $\mathcal{M}\#\mathcal{D} \not\equiv \mathcal{D}\#\mathcal{M}$.

- Grafting is *not necessarily a descent method*. Even if $\mathcal{M}, \mathcal{D}$ are guaranteed to make progress on a deterministic objective (i.e. when $g_t$ is $\nabla F(w_t)$), $\mathcal{M}\#\mathcal{D}$ is not. For instance, if $\mathcal{M}, \mathcal{D}$ are gradient descent with different preconditioners, there is no guarantee that $\mathcal{M}\#\mathcal{D}$ converges, or ever makes progress on $F(w)$.

Grafting can be used as a diagnostic tool to answer the question posed at the end of Section 2, by letting us observe the optimization trajectory and end-to-end performance *when $\mathcal{M}$ and $\mathcal{D}$ are forced to have the same overall learning rate schedule*. This gives a controlled way to study whether $H_t$ or $\eta_t$ is more important in determining an optimizer's empirical success.

---

**Algorithm 1** Grafted meta-optimizer $\mathcal{M}\#\mathcal{D}$

---

1: **init:** optimizers $\mathcal{M}, \mathcal{D}$; hyperparameters for $\mathcal{M}, \mathcal{D}$; partition $\mathcal{P}$ of model parameters.
2: Initialize $\mathcal{M}, \mathcal{D}$ with their hyperparameters.
3: **for** each iteration $t$ **do**
4:     Receive input $(w_t, g_t)$.
5:     Query steps from inner optimizers:
        $w_{\mathcal{M}} \leftarrow \mathcal{M}(w_t, g_t), \ w_{\mathcal{D}} \leftarrow \mathcal{D}(w_t, g_t)$.
6:     **for** each parameter group $\rho \in \mathcal{P}$ **do**
7:         Compute grafted update:

$$w_{\#}^{(\rho)} \leftarrow w_t^{(\rho)} + \frac{\left\| w_{\mathcal{M}}^{(\rho)} - w_t^{(\rho)} \right\|}{\left\| w_{\mathcal{D}}^{(\rho)} - w_t^{(\rho)} \right\|} \left( w_{\mathcal{D}}^{(\rho)} - w_t^{(\rho)} \right).$$

        (Or, if $w_{\mathcal{D}}^{(\rho)} = w_t^{(\rho)}$, let $w_{\#}^{(\rho)} \leftarrow w_t^{(\rho)}$.)
8:     Output grafted weight update:
        $(\mathcal{M}\#\mathcal{D})(w_t, g_t) := w_{\#}$.

---

## 3.1 GRANULARITY OF GRAFTING

Before proceeding to the experiments, we will introduce one more dimension to the grafting procedure: its granularity. Formally, let $\mathcal{P}$ be a partition of $\{1, \ldots, d\}$, supplied as a hyperparameter. For $v \in \mathbb{R}^d, \rho \in \mathcal{P}$, we denote by $v^{(\rho)} \in \mathbb{R}^{|\rho|}$ the restriction of $v$ to the coordinates indexed at $\rho$. Then, we transfer norms separately for each set of coordinates in the partition; this is the complete method summarized by Algorithm 1. Let us note some special cases of $\mathcal{P}$, and their associated properties and interpretations:

- If $\mathcal{P} = \{\{1, \ldots, d\}\}$, then we can think of $\mathcal{M}\#\mathcal{D}$ as transferring an overall learning rate schedule from $\mathcal{M}$ to $\mathcal{D}$. This schedule is not in general a static sequence of scalars $\{\eta_t\}$, and depends on the trajectory of inputs $(w_t, g_t)$. We call this instantiation *global grafting*.

- If we choose instead the most granular partition $\mathcal{P} = \{\{1\}, \ldots, \{d\}\}$, grafting transfers one learning rate schedule per parameter. If the optimizers differ only by a positive diagonal preconditioner (like SGD and AdaGrad), so that the steps have the same sign pattern, then it holds that $\mathcal{M}\#\mathcal{D} \equiv \mathcal{M}$.

- Interpolating between these two extremes, we can set the elements of $\mathcal{P}$ to be the (usually tensor-shaped) parameter groups in the model's implementation. This is a natural partition provided by the optimizer interfaces of all commonly-used deep learning frameworks. We call this variant *layer-wise grafting*, which transfers one learning rate schedule per parameter group from $\mathcal{M}$ to $\mathcal{D}$.

This provides a generalization of the schedule transfer methodology: for any choice of partition, we can force $\mathcal{M}$ and $\mathcal{D}$ to have the same learning rate schedule simultaneously over the parameter groups. By making this partition more granular, we obtain more expressive families of learning rate corrections.

## 4 GRAFTING FOR IMPLICIT SCHEDULE TRANSFER

We first present an empirical study on the transfer of implicit step size schedules between optimizers, on state-of-the-art training benchmarks. Starting with a well-tuned baseline $\mathcal{M}$, our protocol compares two optimizer hyperparameter searches with the *same computational budget*: an optimizer $\mathcal{D}$ which is less frequently used and has worse performance than $\mathcal{M}$ on the corresponding architecture, and $\mathcal{M}\#\mathcal{D}$, fixing the tuned hyperparameters of $\mathcal{M}$. Note that the set of hyperparameters tuned in both cases are also the same i.e. the hyperparameters of $\mathcal{D}$. Under the same hyperparameter tuning protocol and budget we consistently found across architectures/tasks and batch sizes, that grafting induced positive transfer of end-to-end model performance, i.e. the performance of $\mathcal{D}\#\mathcal{M}$ closes the gap (sometimes fully) between the performance $\mathcal{D}$ and $\mathcal{M}$.

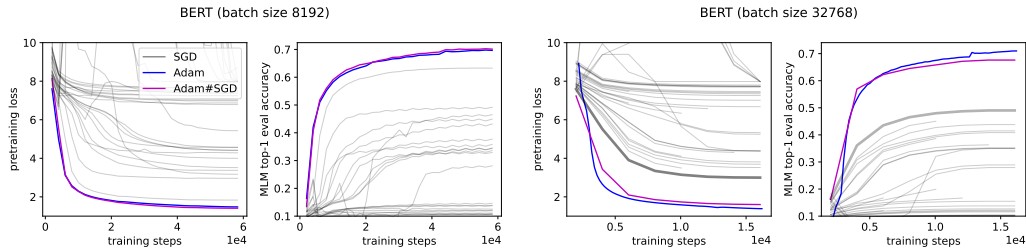

Figure 2: BERT experiments for implicit hyperparameter transfer, comparing hyperparameter search for SGD (with momentum) vs. grafting with $\mathcal{M}$ = Adam. Adam#SGD outperformed all pure SGD runs significantly; most hyperparameter settings for SGD caused training to diverge. Trials on batch sizes 8192 and 32768 are shown.

**BERT pretraining.**   As a timely choice of a resource-intensive training task, we ran this hyperparameter transfer experiment for a BERT (Devlin et al., 2018) model pretraining setup. We make two choices of $\mathcal{M}$, Adam and LAMB (You et al., 2019) commonly reported as the state-of-the-art optimizer for this problem (also see Nado et al. (2021). For $\mathcal{D}$ we choose SGD with momentum, which has been widely reported to be unsuccessful in reaching state-of-the-art performance. In particular, You et al. (2019) explicitly point out in their abstract that their LAMB algorithm is motivated by the fact that LARS (You et al., 2017), its non-adaptively-preconditioned predecessor, performs poorly on self-attention models. We repeated the experiment at batch sizes 8192 (measuring final performance at 56000 training steps) and 32768 (14063 steps). Further details of the hyperparameter search space as well as the full hyperparameter tables are supplied in the appendix.

The model contains 24 transformer layers with 1024 hidden dimensions and 16 attention heads with a total of 340M parameters. It was trained on the combined Wikipedia and Books corpus (Zhu et al., 2015) datasets (2.5B and 800M words, respectively). We used sequence length of 128. We follow the tuning protocol from Nado et al. (2021) that uses a quasi-random search (Bousquet et al., 2017) with a simple search space. Hyperparameters included are learning rate $\eta$, the moment parameters $\beta_1$, $\beta_2$, the polynomial power for the learning rate warmup $p_{warmup}$, and weight decay $\lambda$. We fixed the $\epsilon$ in Adam to $1e-11$ for all BERT experiments. We selected the best trial using the masked language model accuracy over 10k examples from the training set. Further details are provided in the appendix.

Figure 2 and Table 1 summarize our findings. We were able to corroborate the reported difficulty of training Transformers with SGD, with SGD leading to significantly worse performance as compared to Adam or LAMB. On the other hand, we found that Adam#SGD (at the layer-wise granularity) was able to closely match the performance of Adam, even exceeding it at the smaller batch size. This establishes that the performance gap between the untuned optimizers in this setting arises not from Adam's per-coordinate preconditioning, but at the granularity of layer-wise implicit step size schedules. Zhang et al. (2019) propose a theoretical mechanism towards explaining the apparent necessity of adaptive methods for training self-attention models; here, grafting serves as an ablation tool, showing successful convergence with a less granular adaptive preconditioning rule.

| Algorithm | BERT MLM top-1 accuracy | |
|---|---|---|
| | Batch size 8192 | Batch size 32768 |
| Adam | 69.5 | 70.9 |
| SGD | 63.3 | 48.3 |
| Adam#SGD (Layer-wise) | 70.1 | 67.6 |
| LAMB | 70.4 | - |
| LAMB#SGD (Layer-wise) | 71.0 | - |

Table 1: Final Top-1 masked language model validation accuracies for BERT pretraining with Adam, Lamb at 8k and 32K batch size. The global version of grafting were found to perform significantly worse and have been omitted. All the results were repeated multiple times with random seeds and the results are consistent upto a 1% deviation

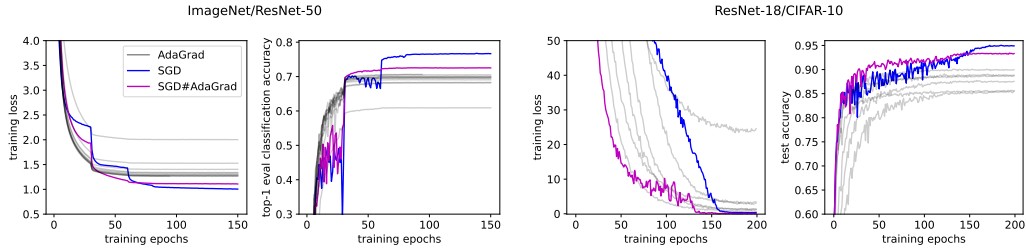

Figure 3: ResNet experiments for implicit hyperparameter transfer, comparing hyperparameter search for AdaGrad vs. grafting with a well-tuned baseline $\mathcal{M}$ = SGD. SGD#AdaGrad outperformed all pure AdaGrad runs. The batch size for the ImageNet run was 8192 and the CIFAR-10 run was 128. Further details can be found in the appendix.

**ImageNet/CIFAR-10 classification with ResNets.** As an alternative setting for grafting, we examine the behavior of grafting on convolutional architectures for image classification tasks: ImageNet (a ResNet-50 with $26M$ parameters) and CIFAR-10 (a ResNet-18 model with about $11M$ parameters). In this setting, we make two choices of $\mathcal{M}$, SGD (with momentum) as well as Adam, with tuned baseline setups derived from Goyal et al. (2017) (ImageNet) and Loshchilov & Hutter (2016) (CIFAR-10). A full description of the hyperparameters is deferred to the appendix. For $\mathcal{D}$ we choose AdaGrad, which is commonly believed to lead to low performance on these models. For these models we observe that the global version of grafting is as effective as the local version and both lead to performance improvements over the Adagrad baseline for these models closing the gap between performance. The results are summarized in Figure 3 and the ImageNet results are summarized in Table 2. We conducted these experiments at multiple regimes of batch sizes for both models and observed consistent trends. The precise numbers for the CIFAR-10 study can be found in the appendix.

It has been noted many times (Zeiler, 2012; Wilson et al., 2017; Bottou et al., 2018) that the AdaGrad optimizer, with commonly used hyperparameter search spaces, fails to train state-of-the-art vision models to the same performance as other, closely related optimizers. These experiments show that the gap can be partially closed with a global learning rate schedule transferred from a successful run of SGD/Adam.

| Algorithm | ImageNet Top-1 Accuracy | |
|:---:|:---:|:---:|
| | Batch size 1024 | Batch size 8192 |
| SGD | 76.85 | 76.68 |
| AdaGrad | 72.93 | 70.63 |
| SGD#AdaGrad (global) | 74.43 | 72.53 |
| SGD#AdaGrad (layer-wise) | 73.85 | 72.59 |
| Adam | 76.47 | 76.27 |
| Adam#AdaGrad (global) | 73.61 | 72.09 |
| Adam#AdaGrad (layer-wise) | 73.42 | 72.45 |

Table 2: Top-1 accuracy at 150 training epochs for ImageNet experiments across batch sizes. The accuracy stabilized at 90 epochs. For all algorithms we performed an extensive tuning over hyperparameters and used the step-decay schedule. SGD#AdaGrad and Adam#AdaGrad are the grafted versions. All the results were repeated multiple times with random seeds and the results are consistent upto a 0.3% deviation.

## 5 GRAFTING FOR EXPLICIT SCHEDULE DISCOVERY

A natural question, in light of the results in Section 4, is whether the benefits of grafting can be distilled into a non-adaptive correction to the algorithm $\mathcal{D}$, eliminating the need to run $\mathcal{M}$ in parallel. In this section, we demonstrate that this pipeline can be completed in both of the large-scale setups from the previous section. We extract the step size ratios $\|w_{\mathcal{M}} - w_t\| / \|w_{\mathcal{D}} - w_t\|$ from a successful execution of grafting, and using these sequences as learning rate schedules for $\mathcal{D}$ in the

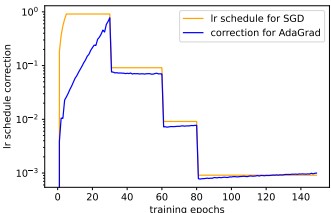

Figure 4: A learning rate schedule for AdaGrad discovered using grafting. Before the first learning rate decay (imposed by the tuned schedule for $\mathcal{M}$=SGD), grafting discovers an implicit polynomial warmup when transferring the performance to $\mathcal{D}$=AdaGrad.

traditional non-adaptive. Applying this form of *transfer* from global grafting results in the discovery of novel learning rate schedules in the usual sense: global, time-dependent, non-adaptive multipliers for the learning rate $\eta_t$. Applied more granularly to layer-wise grafting, this allows for the discovery of a higher-dimensional correction.

**Training a ResNet with AdaGrad.** Using the *global* variant of Algorithm 1 with $(\mathcal{M}, \mathcal{D}) = $ (SGD, AdaGrad) and keeping track of the norms of the steps produced by $\mathcal{M}$ and $\mathcal{D}$ periodically at every epoch, we arrive at a polynomial warmup-like correction overlaid on the stagewise-constant annealing (Goyal et al., 2017) schedule employed by SGD. Figure 4 displays this discovered learning rate schedule compared to the one used on SGD. The final training performance was a top-1 accuracy was 72.46%.

**Training BERT with SGD.** We demonstrate that the protocol of learning rate schedule discovery can be done with per-layer grafting as well. Strikingly, this enables us to discover a simple per-layer step size correction that allows the standard SGD(with momentum) to train a Transformer model without adaptive preconditioning for the first time to the best of our knowledge. In contrast to global grafting which transferred a non-adaptive global learning rate schedule, herein we only transfer a single per-layer scalar correction which is held constant through training.

In particular, we performed the protocol of layer-size grafting as discussed in the previous section, and recorded the per-layer ratio of the norms. Figure 5 provides a collection of some representative ratios. Table 5 and Figure 8 in the appendix provide all the ratios as well as additional visualizations at a finer granularity.

We computed the median of these individual corrections for the first 2000 steps and applied this layer-wise multiplier along with SGD. We note that these corrections effectively rescale the relative learning rate between the layers. We find that this offline transfer is sufficient to push the test accuracy of SGD to the state-of-the-art Adam baseline achieving eval accuracy of 69.5. Full experimental details with can be found in the supplementary material.

**Simplifying the discovered schedule.** The above discussion in particular highlights that standard SGD with momentum can be used to train BERT to state-of-the-art performance, as long as an appropriate per-layer learning rate correction is provided. In this section we explore whether the space of these corrections can be further simplified. This highlights the robustness of our transfer approach, showing that performance is preserved as long as relative per-layer scales are approximately preserved and provides a proof of concept that search space for these corrections is significantly smaller than the number of parameter groups. We only consider the case of batch-size 8192 here, similar results are obtained for batch size 32K and the presentation is deferred to the appendix.

To provide a proof-of-concept, motivated by trends observed in Figure 5, we tie the per-layer scalar corrections for all the parameter groups in the attention layers to a fixed value $\beta$, and simply discretize the obtained medians for the other parameter groups/ layers (eg. softmax and embeddings) to the nearest power of 10 which is smaller. This results in a very simple scalar correction scheme which has values in the set $\mathcal{D}_\beta = \{0.1, 1, 10, 100, \beta\}$. We provide the achieved accuracy results via the above corrections in Table 6b over some choices of $\beta$. We find that performance tends to be relatively stable to small changes in choices of $\beta$. We stress that our aim is to demonstrate a

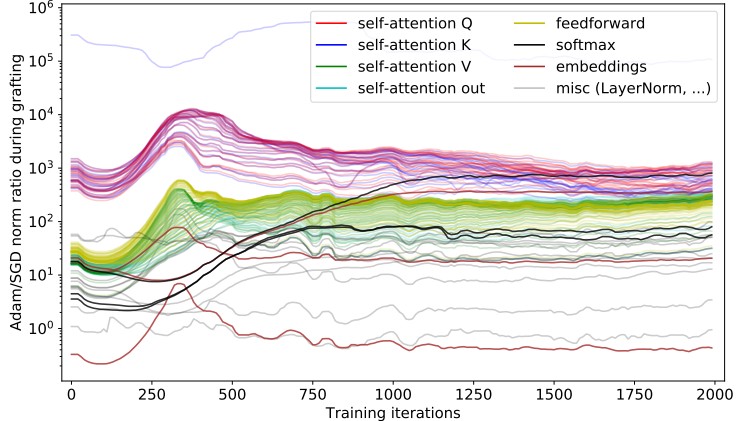

Figure 5: Per-layer learning rate corrections during the first 2000 iterations of BERT pretraining with Adam#SGD, enabling the discovery of the learning rate correction to SGD. More visualizations are provided in the supplementary material.

| Algorithm | Eval accuracy |
|---|---|
| Adam | 69.5 |
| SGD | 63.3 |
| Adam#SGD | 70.1 |
| SGD (Medians) | 69.5 |
| SGD ($\mathcal{D}_{500}$) | 70.0 |
| SGD ($\mathcal{D}_{250}$) | 68.9 |
| SGD ($\mathcal{D}_{50}$) | 67.8 |

(a) Final top-1 validation accuracies for BERT pretraining with discovered per-layer learning rate corrections.

| Parameter group | Median | Rounded ($\mathcal{D}_{500}$) |
|---|---|---|
| Position embeddings | 18.8 | 10 |
| Word embeddings | 312.9 | 100 |
| Layer 1 $Q$ | 525.5 | 500 |
| Layer 1 $K$ | 466.6 | 500 |
| Layer 1 $V$ | 312.9 | 500 |
| Layer 24 $Q$ | 1090.1 | 500 |
| Layer 24 $K$ | 1081.3 | 500 |
| Layer 24 $V$ | 193.9 | 500 |
| Softmax | 62.8 | 10 |

(b) Some representative per-layer learning rate corrections discovered using grafting, then simplified manually. These corrections, which can be viewed as a 168-dimensional hyperparameter, enable SGD to pretrain a BERT with performance competitive with adaptive methods.

proof-of-concept that a simplified correction scheme (in this case composed of only 5 distinct values) works and thus to shed a light on the extent of the role of adaptivity in training these models. A careful study of the space of all possible discretizations is out of the scope and if left for future work.

## 6 Conclusion

We have introduced learning rate grafting, a binary operation which blends the behavior of two optimization algorithms, towards investigating the entanglements between widely-used adaptive preconditioning rules and learning rate schedules. We have presented an empirical study with popular optimizers for state-of-the-art deep architectures, discovering that a well-performing optimizer $\mathcal{M}$ can transfer its performance to $\mathcal{D}$ via grafting its sequence of implicit step size schedules. Furthermore, we have shown that grafting can be used to extract standalone learning rate corrections, enabling us to train a Transformer using SGD (with momentum) for the first time. We hope that this finding will stimulate further empirical research on the power of simple per-layer learning rate schedules.

The empirical phenomena examined in this work seem to be unexplained by current theory. We hope that the experiments enabled by grafting will aid in developing more robust beliefs about both adaptive methods and learning rate schedules, and guide future theoretical inquiry.

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

## A  ADDITIONAL EXPERIMENTS AND DETAILS

We provide additional experiments and all the implementation details in this section. In the next section we provide full hyper-parameter search tables for all our experiments to aid reproduction of our experiments.

### A.1  BERT EXPERIMENTS

#### A.1.1  IMPLEMENTATION DETAILS

We used the same experimental setup as the official BERT codebase[1] and the standard train/test split from the previous literature. We trained on Google TPUs, using TPUv3-128 / TPUv3-256 for the 8K and 32K batch size experiments.

We trained the two pretraining objectives on the combined Wikipedia and Books corpus (Zhu et al., 2015) datasets (2.5B and 800M words, respectively). We used sequence lengths of 128.

Our experimentation protocol is identical to tyat described in Nado et al. (2021): we tuned hyperparameters using quasi-random search (Bousquet et al., 2017) with a fixed budget in a simple search space described in Table A.1.1. We selected the best trial using the masked language model accuracy over 10k examples from the training set. The experiments were run for a total of 56352 steps for batch size 8K and 14063 steps for batch size 32K.

**Adam.** Hyperparameters included learning rate $\eta$, exponential window parameters $\beta_1$, $\beta_2$, the polynomial power for the learning rate warmup and decay $p$, and weight decay $\lambda$. We fixed the $\epsilon$ in Adam to $10^{-11}$ for all BERT experiments. Polynomial power was searched over $\{1, 2\}$ for batch size 8K and set to 1 for 32K. The entire hyperparameter search and associated performance can be found in Tables 7 and 11 for batch sizes 8K and 32K respectively.

**SGD with momentum.** Hyperparameters included learning rate $\eta$, $\beta_1$, the polynomial power for the learning rate warmup and decay $p$, and weight decay $\lambda$. Polynomial power was searched over 1,2 for batch size 8K and set to 1 for 32K. The entire hyperparameter search and associated performance can be found in Tables 8 and 12 for batch sizes 8K and 32K respectively.

**Grafting: Adam#SGD.** We use the above described implementations of Adam and SGD with momentum as $\mathcal{M}, \mathcal{D}$ respectively. The Adam hyperparameters were fixed to the optimal ones found. The SGD hyperparameters were tuned with the same protocol as for SGD alone. The overall learning rate scalar was also tuned. Tables 9 and 13 contain the hyperparameter search and associated performance for batch sizes 8K and 32K respectively.

---

[1] https://github.com/google-research/bert

| Hyperparameter | Range | Scaling |
|---|---|---|
| $p$ | $\{1, 2\}$ | discrete |
| $\eta$ | $[10^{-5}, 1.0]$ | logarithmic |
| $1 - \beta_1$ | $[10^{-3}, 0.5]$ | logarithmic |
| $1 - \beta_2$ | $[10^{-3}, 0.5]$ | logarithmic |
| $\lambda$ | $[10^{-6}, 10^{-3}]$ | logarithmic |

Table 3: The search space used to tune Adam/SGD on BERT. $\lambda$ refers to weight decay and $p$ refers to the polynomial power in the learning rate schedule for both the warmup and decay phases.

**LAMB** - We use the exact same settings as in the original paper (You et al., 2019).

**Grafting: LAMB#SGD.** We use the above described implementations of LAMB and SGD with momentum as $\mathcal{M}, \mathcal{D}$ respectively. The LAMB hyperparameters were fixed to the optimal ones. The SGD hyperparameters were tuned with the same protocol as for SGD alone. The overall learning rate scalar was also tuned. Table 10 contains the hyperparameter search and associated performance for batch sizes 8K and 32K respectively.

A.1.2 TRAINING BERT WITH SGD (BATCH SIZE 8K AND 32K)

Some of these results are partly presented in the main paper. We provide a full exposition for completeness.

We demonstrate that the protocol of learning rate schedule discovery can be done with per-layer grafting as well. Strikingly, this enables us to discover a simple per-layer step size correction that allows the standard SGD(with momentum) to train a Transformer model without adaptive preconditioning for the first time to the best of our knowledge. In contrast to global grafting which transferred a non-adaptive global learning rate schedule, herein we only transfer a single per-layer scalar correction which is held constant through training.

In particular, we performed the protocol of layer-size grafting as discussed in the previous section, and recorded the per-layer ratio of the norms. Figure 7 shows these ratios for batch sizes 8K and 32K. Given the apparent instability of the ratios on the later part of the runs for the 8K, we zoom in on the initial part of the training.[2] In particular, we computed the median of these individual corrections for the first 2000 (respectively 500) steps for batch size 8K (respectively 32K) and applied this layer-wise multiplier along with SGD (with momentum); see Figures 6 and 8 for visualizations of the schedule corrections, and Table 5 for the median values. We find that this offline transfer is sufficient to push the test accuracy of SGD much closer to the state-of-the-art Adam baseline (see Table 4).

| Algorithm | Masked LM Accuracy | |
|---|---|---|
| | Batch size 8K | Batch size 32K |
| Adam | 69.5 | 70.8 |
| SGD | 63.3 | 49.0 |
| Adam#SGD | 70.1 | 67.6 |
| SGD (Medians) | 69.5 | 63.8 |

Table 4: Final top-1 validation accuracies for BERT pretraining with Adam, SGD(with momentum), Adam#SGD and the discovered per-layer learning rate corrections via medians.

---

[2]Despite these exponentially large correction ratios, the grafted optimizer converges successfully. We leave an investigation of this curious phenomenon for future work.

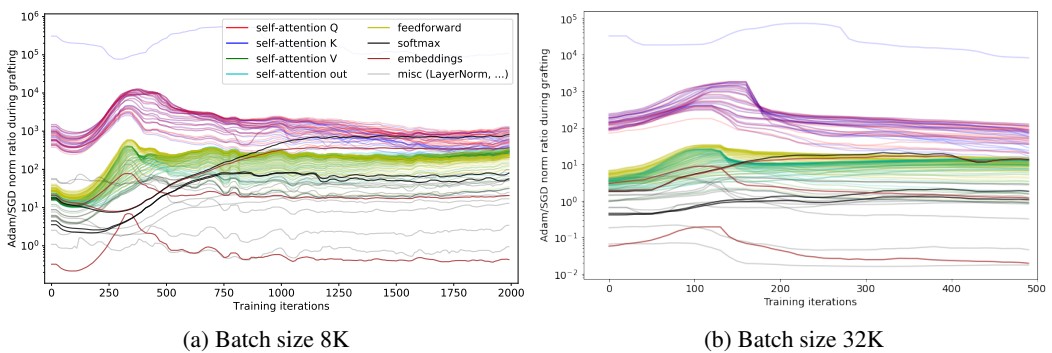

(a) Batch size 8K

(b) Batch size 32K

Figure 6: Per-layer learning rate corrections during the first 2000 (resp. 500) iterations of BERT pretraining with Adam#SGD with batch size 8K (resp. 32K), enabling the discovery of the learning rate correction to SGD. A median filter of width 51 is applied to each sequence, for clarity.

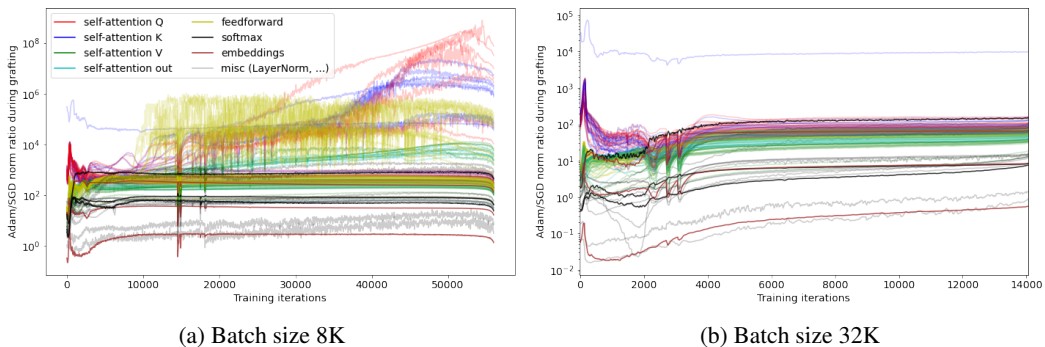

(a) Batch size 8K

(b) Batch size 32K

Figure 7: Per-layer learning rate corrections for all training iterations, in the BERT pretraining setups with Adam#SGD. Some of these corrections increase exponentially; note that despite performing these large corrections, the grafted optimizer converges to nearly state-of-the-art accuracy.

| Parameter group | Median correction (8K) | Median correction (32K) |
|---|---|---|
| Embedding LN $\beta$ | 9.10 | 0.54 |
| Embedding LN $\gamma$ | 16.34 | 2.26 |
| Position embeddings | 18.83 | 2.48 |
| Token type embeddings | 0.45 | 0.03 |
| Word embeddings | 312.94 | 7.94 |
| Attention $Q$ bias | 378.07 | 38.56 |
| Attention $K$ bias | 154101.58 | 25119.54 |
| Attention $V$ bias | 37.69 | 2.01 |
| Attention out bias | 21.92 | 1.16 |
| Attention FC1 bias | 81.30 | 4.57 |
| Attention FC2 bias | 23.87 | 1.24 |
| Attention LN1 $\beta$ | 21.60 | 1.13 |
| Attention LN1 $\gamma$ | 41.99 | 4.08 |
| Attention LN2 $\beta$ | 22.05 | 1.12 |
| Attention LN2 $\gamma$ | 45.42 | 3.83 |
| Pooler bias | 32.69 | 0.64 |
| Pooler weights | 48.73 | 1.02 |
| MLM output bias | 479.54 | 8.74 |
| MLM LN $\beta$ | 21.78 | 0.57 |
| MLM LN $\gamma$ | 17.19 | 0.82 |
| MLM pre-LN bias | 48.04 | 0.71 |
| MLM pre-LN weights | 62.77 | 1.06 |
| NSP bias | 0.66 | 0.01 |
| NSP weights | 2.00 | 0.04 |
| Attention $Q$ weights (layers 1-24) | {525.50, 575.42, 714.81, 794.62, 900.46, 904.97, 1150.11, 1290.34, 1411.08, 1867.67, 2074.47, 1975.08, 859.45, 2026.33, 1739.56, 1549.03, 1602.61, 1516.13, 1291.10, 1101.66, 1124.54, 850.50, 993.93, 1090.10} | {39.00, 80.19, 103.98, 104.84, 146.52, 202.22, 187.58, 229.54, 279.68, 236.02, 267.71, 272.56, 255.77, 227.56, 216.35, 207.21, 194.48, 172.97, 150.94, 145.22, 108.78, 84.44, 90.25, 69.70} |
| Attention $K$ weights (layers 1-24) | {466.59, 503.57, 662.77, 679.56, 764.49, 714.15, 958.21, 1086.80, 1266.25, 1731.75, 1980.33, 1842.08, 595.65, 1949.40, 1669.08, 1335.04, 1495.53, 1452.81, 1098.97, 1027.76, 1096.50, 709.72, 838.97, 1081.26} | {34.70, 66.95, 98.20, 90.28, 123.40, 185.35, 180.04, 227.60, 265.37, 206.99, 262.27, 276.03, 253.69, 231.60, 227.34, 211.37, 205.17, 179.20, 155.37, 142.24, 101.69, 72.56, 77.83, 64.13} |
| Attention FC1 weights (layers 1-24) | {82.02, 108.41, 131.46, 151.14, 169.54, 186.14, 198.20, 208.18, 217.43, 228.15, 230.22, 228.92, 237.99, 239.71, 236.57, 235.85, 231.04, 224.10, 219.76, 214.12, 206.27, 203.64, 207.25, 216.04} | {7.17, 11.03, 14.30, 16.96, 18.66, 19.93, 20.77, 21.24, 21.95, 22.31, 22.27, 21.70, 21.41, 21.04, 20.27, 19.43, 18.38, 17.47, 16.04, 14.29, 12.70, 10.49, 8.66, 6.24} |
| Attention $V$ weights (layers 1-24) | {48.61, 66.94, 85.83, 96.88, 112.34, 129.41, 135.96, 150.30, 167.87, 184.03, 203.72, 202.35, 178.28, 214.61, 210.79, 209.75, 216.17, 226.42, 221.07, 220.83, 219.30, 197.48, 187.25, 193.95} | {2.58, 4.56, 7.23, 9.69, 10.97, 12.42, 13.39, 13.25, 14.18, 14.48, 14.72, 14.27, 14.25, 14.16, 13.65, 12.89, 12.26, 11.58, 10.93, 9.85, 8.55, 7.40, 6.32, 4.71} |
| Attention out weights (layers 1-24) | {56.17, 74.65, 97.39, 105.54, 120.98, 137.54, 145.35, 167.26, 186.15, 204.79, 212.83, 212.14, 202.61, 232.44, 222.43, 228.23, 228.55, 238.57, 234.49, 232.85, 241.87, 229.22, 222.86, 213.22} | {2.92, 5.59, 8.01, 10.51, 12.55, 13.42, 13.61, 14.52, 14.51, 14.45, 14.40, 14.44, 14.49, 14.41, 13.88, 13.13, 12.61, 11.76, 10.89, 9.91, 8.84, 7.40, 6.35, 4.65} |
| Attention FC2 weights (layers 1-24) | {71.84, 88.45, 103.35, 115.39, 126.44, 137.14, 148.70, 159.31, 167.05, 179.80, 181.04, 183.60, 192.79, 200.97, 201.98, 207.50, 208.11, 206.15, 208.88, 210.20, 207.35, 209.14, 216.43, 218.14} | {5.56, 7.32, 8.64, 9.54, 10.34, 11.12, 12.05, 12.71, 13.44, 13.89, 14.37, 14.74, 14.83, 14.89, 15.00, 14.69, 14.14, 13.83, 13.19, 12.05, 10.81, 9.16, 7.38, 5.26} |

Table 5: Corrections for each parameter group for the BERT schedule discovery experiments, found using grafting but executed indepedently as a correction on SGD.

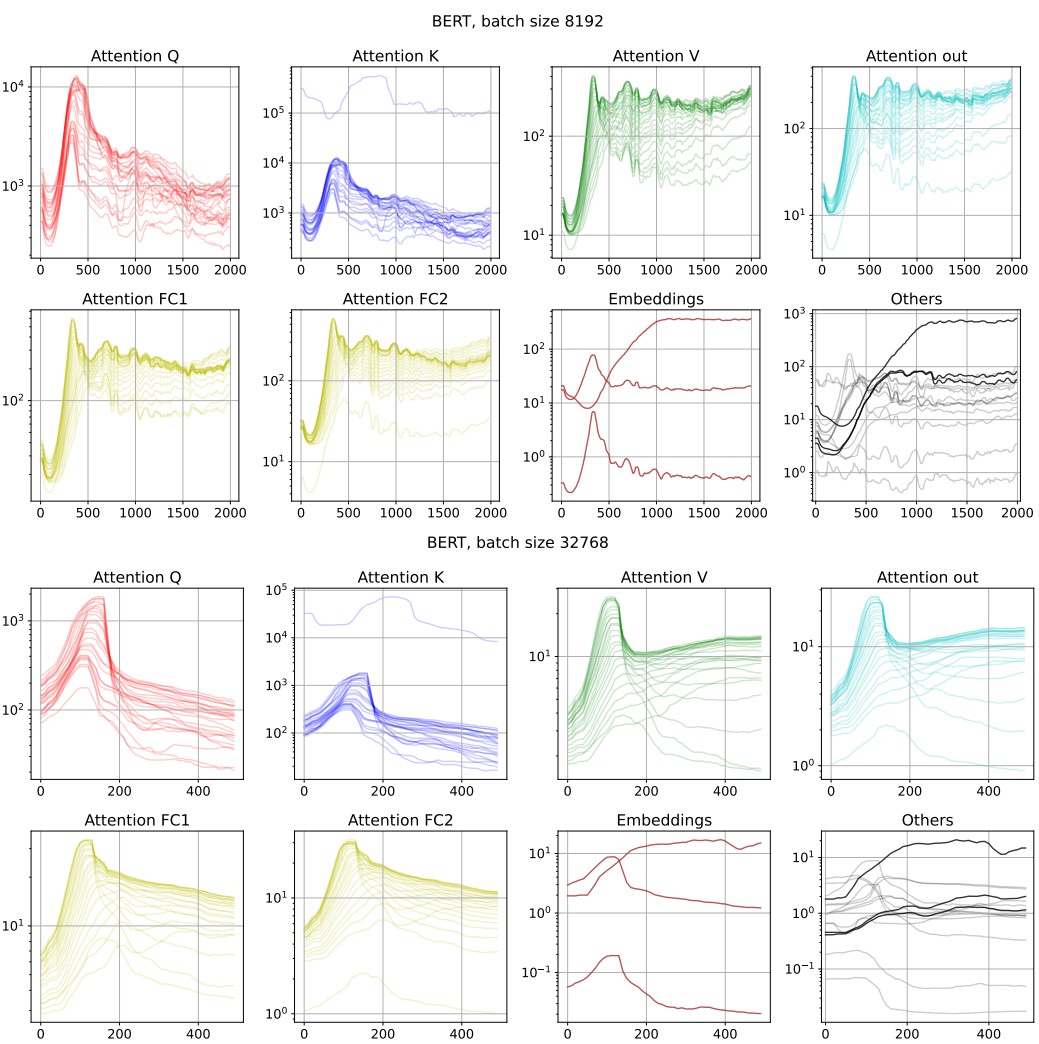

Figure 8: Per-layer learning rate schedule corrections from Figure 6, displayed on separate plots by type of parameter group.

## A.2 IMAGENET EXPERIMENTS

### A.2.1 IMPLEMENTATION DETAILS

For the image classification task with Imagenet we use the standard Resnet-50 model He et al. (2016). The model code, training pipeline and other parameters are identical to the MLPerf training benchmark for ResNet-50 v1.5 on ImageNet Mattson et al. (2019). All the training experiments are carried out up to 150 epochs (however all training stabilizes around 90 epochs). We report the achieved test top-1 accuracy. For all the cases we implement a distributed version of batch-norm akin to the one used in Goyal et al. (2017). Each core locally computes and maintains batch-norm statistics over sub-batches of 64 (akin to the $n$ parameter in Goyal et al. (2017)). The maintained batch norm statistics are synchronized at the end of every epoch. The batch norm momentum is set to 0.9 for batch size 8192 and 0.99 for batch size 1024 and epsilon is set to 0.00001. The experiments are performed on a 16 TPU core v3 (for batch size 1024) and a 64 TPU core v3 (for batch size 8192). The timings for a single run are approximately 1.5 hours and 35 minutes for batch size 1024 and 8192 respectively.

**SGD with momentum.** We use the standard implementation of SGD with Heavy Ball Momentum with the momentum parameter set to 0.9 and the weight decay set to 0.0001. We use the warmup+step decay schedule proposed in Goyal et al. (2017). The schedule warms up the learning rate linearly to the *base learning rate* for 5 epochs and then the learning rate is held constant until cut by a factor of 0.1 at epoch 30,50,80. Table 15 contains the hyperparameter tuning information where the search was performed over the base learning rate and across two batch sizes.

**Adam.** We use the standard implementation of Adam and tune the momentum parameter and the weight decay. We fixed $\beta_2$ to the default 0.999. We use the warmup+step decay schedule proposed in Goyal et al. (2017). The schedule warms up the learning rate linearly to the *base learning rate* for 5 epochs and then the learning rate is held constant until cut by a factor of 0.1 at epoch 30,50,80. Tables 16 and 17 contain the hyperparameter tuning information where the search was performed over the base learning rate and across two batch sizes.

**AdaGrad.** We use the standard implementation of AdaGrad (present in TensorFlow). We use all the default hyperparameters except the learning rate which we search over. We use the same learning rate schedule as reported in SGD with Momentum (step-decay). We highlight that this schedule worked significantly better than no schedule at all which is traditionally what has been reported for AdaGrad baselines leading to the stronger baselines for AdaGrad baselines we present. Table 14 contains the hyperparameter tuning information where the search was performed over the base learning rate and across two batch sizes.

**Grafting: SGD#AdaGrad/Adam#Adagrad.** We use the above described implementations of SGD with momentum/Adam and AdaGrad as $\mathcal{M}, \mathcal{D}$ respectively. Since the only hyper-parameter for Adagrad is a learning rate , we tune the base learning rate supplied to SGD (as the $\mathcal{M}$ algorithm governs the overall learning rate of the grafted algorithm) Table 18 contains the hyperparameter tuning information where the search was performed over the base learning rate and across two batch sizes.

## A.3 CIFAR-10 EXPERIMENTS

For the image classfication task with CIFAR-10 dataset we use a standard PreActivation Resnet-18 model (He et al., 2016). The model code is implemented identically as in the following repository (kuangliu, 2017). All the training experiments are carried out upto 200 epochs. We report the achieved test accuracy. The experiments are performed on a single Tesla V100-SXM2 GPU. The timings for a single run are approximately 90 minutes.

**SGD with Momentum.** We use the standard implementation of SGD with Heavy Ball Momentum with the momentum parameter set to 0.9 and the weight decay set to 0.0005. We use the single cycle cosine decay learning rate schedule suggested by Loshchilov & Hutter (2016). Formally the learning

rate at the start of epoch $t$ given the base learning rate $\beta$ is given by

$$\beta \cdot \cos(\pi/2(t/200))$$

Table 20 contains the hyperparameter tuning information where the search was performed over the base learning rate and across three batch sizes.

**AdaGrad.** We use the standard implementation of AdaGrad (present in TensorFlow). We use all the default hyperparameters except the learning rate which we search over. We use the same cosine schedule as reported in SGD with Momentum as we found it to be better than applying no schedule at all. Table 19 contains the hyperparameter tuning information where the search was performed over the base learning rate and across three batch sizes.

**Grafting: SGD#AdaGrad.** We use the above described implementations of SGD with momentum and AdaGrad as $\mathcal{M}, \mathcal{D}$ respectively. Further we tune the base learning rate supplied to SGD (as the $\mathcal{M}$ algorithm governs the overall learning rate of the grafted algorithm) Table 21 contains the hyperparameter tuning information where the search was performed over the base learning rate and across three batch sizes. It is worth noting that while the additional learning rate tuning leads to small benefits, running grafting with the tuned SGD learning rate already provides a strong baseline.

Table 6 contains the comparisons between the tuned results.

| Algorithm | CIFAR-10 Top-1 Accuracy | | |
|---|---|---|---|
| | Batch size 128 | Batch size 512 | Batch size 2048 |
| SGD | 94.93 | 94.39 | 93.37 |
| AdaGrad | 89.81 | 88.25 | 85.71 |
| SGD#AdaGrad | 93.42 | 92.36 | 89.81 |

Table 6: Top-1 accuracy at 200 training epochs for CIFAR-10 experiments across batch sizes.

# B    FULL HYPERPARAMETER SEARCH TABLES

## B.1    BERT EXPERIMENTS

### B.1.1    BERT (BATCH SIZE 8K)

| Poly Power ($p$) | $1 - \beta_1$ | $1 - \beta_2$ | Weight Decay Rate ($\lambda$) | Learning Rate | Masked LM Accuracy (as fraction) |
|---|---|---|---|---|---|
| 2 | 0.014533 | 0.305531 | 3.469230 | 0.022672 | 0.000000 |
| 1 | 0.019056 | 0.068170 | 0.174876 | 0.027856 | 0.000000 |
| 1 | 0.043181 | 0.418378 | 1.930781 | 0.000571 | 0.000000 |
| 1 | 0.012960 | 0.067712 | 1.286336 | 0.002391 | 0.000000 |
| 2 | 0.396269 | 0.019148 | 0.003123 | 0.087673 | 0.000000 |
| 1 | 0.104732 | 0.493163 | 0.049373 | 0.000430 | 0.000000 |
| 2 | 0.439164 | 0.126745 | 0.009961 | 0.010594 | 0.000000 |
| 1 | 0.012784 | 0.077563 | 0.005028 | 0.002035 | 0.000000 |
| 1 | 0.242170 | 0.097739 | 0.653928 | 0.002798 | 0.033710 |
| 1 | 0.014564 | 0.054001 | 0.002970 | 0.000214 | 0.033710 |
| 1 | 0.116967 | 0.286304 | 4.401929 | 0.006926 | 0.033710 |
| 2 | 0.406820 | 0.072237 | 0.008125 | 0.015782 | 0.048026 |
| 1 | 0.118934 | 0.012215 | 0.473520 | 0.003294 | 0.048026 |
| 1 | 0.124374 | 0.010643 | 4.738276 | 0.002856 | 0.048646 |
| 1 | 0.114625 | 0.129364 | 0.762753 | 0.026747 | 0.057222 |
| 2 | 0.016942 | 0.010706 | 0.010191 | 0.000042 | 0.596373 |
| 2 | 0.018620 | 0.479633 | 0.280896 | 0.000030 | 0.600682 |
| 2 | 0.056560 | 0.025435 | 0.545407 | 0.000050 | 0.602233 |
| 1 | 0.277735 | 0.316840 | 0.003889 | 0.000037 | 0.614416 |
| 1 | 0.051057 | 0.055620 | 0.001138 | 0.000051 | 0.634374 |
| 1 | 0.099721 | 0.473195 | 1.174743 | 0.000412 | 0.667841 |
| 2 | 0.039101 | 0.133753 | 1.472112 | 0.000956 | 0.672648 |
| 2 | 0.235072 | 0.186723 | 0.828703 | 0.000751 | 0.693680 |
| 1 | 0.047688 | 0.246138 | 0.075992 | 0.000151 | 0.696613 |

Table 7: Hyperparameter search space for Adam, BERT, batch size 8K.

| Poly Power ($p$) | $1 - \beta_1$ | Weight Decay Rate ($\lambda$) | Learning Rate | Masked LM Accuracy (as fraction) |
|---|---|---|---|---|
| 2 | 0.015823 | 0.036840 | 0.054733 | 0.000000 |
| 1 | 0.096770 | 0.426730 | 0.056828 | 0.000912 |
| 2 | 0.112533 | 0.382084 | 0.024687 | 0.057222 |
| 1 | 0.153500 | 0.105432 | 0.026578 | 0.057222 |
| 1 | 0.235137 | 4.153353 | 0.056153 | 0.057222 |
| 1 | 0.204847 | 0.104750 | 0.036520 | 0.057222 |
| 1 | 0.011275 | 4.299596 | 0.003757 | 0.057222 |
| 2 | 0.224600 | 1.709058 | 0.052954 | 0.057222 |
| 1 | 0.084911 | 0.002031 | 0.010545 | 0.057222 |
| 2 | 0.046596 | 2.510824 | 0.003590 | 0.057222 |
| 2 | 0.030383 | 1.391879 | 0.002720 | 0.057222 |
| 1 | 0.174070 | 0.033287 | 0.005125 | 0.057299 |
| 2 | 0.017656 | 4.237251 | 0.000495 | 0.087236 |
| 1 | 0.277091 | 0.003713 | 0.000025 | 0.107089 |
| 2 | 0.259132 | 1.357934 | 0.000693 | 0.107096 |
| 1 | 0.255655 | 1.054776 | 0.000041 | 0.107335 |
| 2 | 0.104964 | 1.613354 | 0.000066 | 0.111124 |
| 1 | 0.276399 | 0.011414 | 0.000054 | 0.119279 |
| 2 | 0.022524 | 3.054689 | 0.000019 | 0.121293 |
| 2 | 0.052927 | 3.238563 | 0.000501 | 0.125202 |
| 2 | 0.177523 | 0.002682 | 0.000126 | 0.137419 |
| 2 | 0.311772 | 0.009484 | 0.000209 | 0.138753 |
| 1 | 0.034004 | 0.001888 | 0.000031 | 0.146332 |
| 1 | 0.131274 | 0.942633 | 0.001947 | 0.281134 |
| 2 | 0.105729 | 0.153407 | 0.000419 | 0.328938 |
| 1 | 0.180943 | 0.001704 | 0.000379 | 0.335963 |
| 1 | 0.013722 | 0.222883 | 0.000048 | 0.343942 |
| 1 | 0.165404 | 0.017183 | 0.000395 | 0.345149 |
| 1 | 0.045047 | 0.948157 | 0.000550 | 0.358314 |
| 1 | 0.085101 | 0.060706 | 0.000309 | 0.377149 |
| 1 | 0.157212 | 0.015188 | 0.000701 | 0.431824 |
| 2 | 0.082335 | 0.030545 | 0.000618 | 0.449213 |
| 1 | 0.183162 | 0.024755 | 0.003892 | 0.465375 |
| 2 | 0.011765 | 0.738450 | 0.000429 | 0.491509 |
| 2 | 0.020559 | 0.007351 | 0.001888 | 0.632697 |

Table 8: Hyperparameter search space for SGD, BERT, batch size 8K.

| Poly Power ($p$) | $1 - \beta_1$ | Weight Decay Rate ($\lambda$) | Learning Rate | Masked LM Accuracy (as fraction) |
|---|---|---|---|---|
| 2 | 0.008817 | 1.523654e-03 | 0.247838 | 0.000000 |
| 2 | 0.025831 | 7.554433e-09 | 0.010177 | 0.000000 |
| 2 | 0.004666 | 7.457963e-05 | 0.041305 | 0.000000 |
| 2 | 0.042243 | 5.682363e-08 | 0.161833 | 0.000000 |
| 2 | 0.001035 | 2.020054e-10 | 0.002286 | 0.000000 |
| 2 | 0.034458 | 7.282971e-10 | 0.082530 | 0.002035 |
| 2 | 0.022501 | 1.747059e-06 | 0.005476 | 0.032267 |
| 2 | 0.002092 | 4.521310e-03 | 0.003641 | 0.032267 |
| 2 | 0.017431 | 1.669821e-06 | 0.009074 | 0.048646 |
| 2 | 0.005098 | 7.112310e-03 | 0.142092 | 0.057222 |
| 2 | 0.016812 | 2.641299e-10 | 0.052162 | 0.057222 |
| 2 | 0.009421 | 3.166860e-08 | 0.006412 | 0.057222 |
| 2 | 0.001396 | 4.255182e-05 | 0.120496 | 0.057222 |
| 2 | 0.001615 | 8.588940e-07 | 0.001236 | 0.057222 |
| 2 | 0.013455 | 5.000904e-07 | 0.201592 | 0.057222 |
| 2 | 0.016216 | 8.084465e-03 | 0.034562 | 0.057222 |
| 2 | 0.003546 | 5.341552e-08 | 0.002101 | 0.057222 |
| 2 | 0.048880 | 7.500002e-01 | 0.018438 | 0.100170 |
| 2 | 0.072832 | 3.817323e-01 | 0.033588 | 0.110570 |
| 2 | 0.190460 | 8.796291e-02 | 0.464158 | 0.357879 |
| 2 | 0.349429 | 2.119862e-06 | 0.002255 | 0.672795 |
| 2 | 0.070881 | 1.240433e-08 | 0.001685 | 0.685996 |
| 2 | 0.042185 | 7.085360e-04 | 0.054075 | 0.687168 |
| 2 | 0.120310 | 3.566263e-07 | 0.038694 | 0.691329 |
| 2 | 0.075832 | 4.531688e-05 | 0.015860 | 0.701189 |

Table 9: Graft Bert 8K: Adam#SGD

| Learning Rate | $1 - \beta_1$ | Weight Decay Rate ($\lambda$) | Masked LM Accuracy (as fraction) |
|---|---|---|---|
| 0.557378 | 0.052662 | 0.008343 | 0.000000 |
| 0.000051 | 0.010000 | 0.000100 | 0.000000 |
| 0.010168 | 0.104408 | 0.118838 | 0.057684 |
| 0.000001 | 0.250000 | 1.000000 | 0.082367 |
| 0.000001 | 0.010000 | 0.000100 | 0.108002 |
| 0.000002 | 0.228890 | 0.046535 | 0.109349 |
| 0.000339 | 0.250000 | 1.000000 | 0.134516 |
| 0.001165 | 0.250000 | 1.000000 | 0.140487 |
| 0.001274 | 0.010000 | 0.005104 | 0.296781 |
| 0.000014 | 0.024513 | 0.000535 | 0.309587 |
| 0.000051 | 0.099820 | 0.072073 | 0.346744 |
| 0.001021 | 0.184613 | 0.167212 | 0.415491 |
| 0.000052 | 0.105862 | 0.003519 | 0.431919 |
| 0.001568 | 0.037527 | 1.000000 | 0.439019 |
| 0.000514 | 0.040944 | 0.254123 | 0.503474 |
| 0.000501 | 0.212621 | 0.017731 | 0.529117 |
| 0.001231 | 0.209148 | 0.008710 | 0.599985 |
| 0.000909 | 0.017010 | 0.218740 | 0.601494 |
| 0.000159 | 0.010000 | 0.000100 | 0.631107 |
| 0.000997 | 0.250000 | 0.000100 | 0.642831 |
| 0.000865 | 0.101639 | 0.004944 | 0.647802 |
| 0.001007 | 0.149186 | 0.000757 | 0.657476 |
| 0.001000 | 0.050000 | 0.010000 | 0.669588 |
| 0.000962 | 0.049001 | 0.007475 | 0.676035 |
| 0.000810 | 0.010000 | 0.016944 | 0.688638 |
| 0.001808 | 0.031094 | 0.004114 | 0.710352 |

Table 10: Hyperparameter search space for LAMB#SGD, BERT, batch size 8K.

## B.1.2 BERT (BATCH SIZE 32K)

| Poly Power ($p$) | $1 - \beta_1$ | $1 - \beta_2$ | Weight Decay Rate ($\lambda$) | Learning Rate | Masked LM Accuracy (as fraction) |
|---|---|---|---|---|---|
| 1 | 0.446131 | 0.025372 | 0.107242 | 0.065235 | 0.000000 |
| 1 | 0.010053 | 0.056364 | 0.171185 | 0.000041 | 0.000000 |
| 1 | 0.260209 | 0.311713 | 0.010859 | 0.003011 | 0.000000 |
| 1 | 0.043480 | 0.018414 | 0.003984 | 0.045078 | 0.000000 |
| 1 | 0.116452 | 0.374498 | 0.338193 | 0.057663 | 0.000000 |
| 1 | 0.011695 | 0.063804 | 0.003009 | 0.000230 | 0.000000 |
| 1 | 0.010612 | 0.379281 | 0.010484 | 0.000680 | 0.000000 |
| 1 | 0.049212 | 0.021061 | 0.154526 | 0.005826 | 0.000000 |
| 1 | 0.011473 | 0.010763 | 0.023060 | 0.004987 | 0.000000 |
| 1 | 0.094829 | 0.034333 | 0.007080 | 0.003053 | 0.000000 |
| 1 | 0.182225 | 0.116353 | 0.002412 | 0.005812 | 0.000000 |
| 1 | 0.035019 | 0.297140 | 0.523223 | 0.009278 | 0.000000 |
| 1 | 0.156544 | 0.277251 | 0.662085 | 0.059544 | 0.000000 |
| 1 | 0.016542 | 0.193700 | 0.707171 | 0.021833 | 0.000000 |
| 1 | 0.150077 | 0.286466 | 0.299879 | 0.093842 | 0.000000 |
| 1 | 0.012764 | 0.012950 | 0.692721 | 0.001412 | 0.000029 |
| 1 | 0.016431 | 0.026711 | 0.012575 | 0.001505 | 0.000057 |
| 1 | 0.072118 | 0.017038 | 2.161319 | 0.059588 | 0.029695 |
| 1 | 0.242228 | 0.014281 | 7.335423 | 0.000740 | 0.057367 |
| 1 | 0.014321 | 0.032597 | 7.421835 | 0.000292 | 0.057367 |
| 1 | 0.044166 | 0.023843 | 0.047968 | 0.000014 | 0.429771 |
| 1 | 0.107729 | 0.013147 | 2.653585 | 0.000018 | 0.453644 |
| 1 | 0.334832 | 0.069228 | 0.040715 | 0.000043 | 0.549747 |
| 1 | 0.079745 | 0.019922 | 0.007470 | 0.000053 | 0.564666 |
| 1 | 0.054556 | 0.035450 | 0.005083 | 0.000059 | 0.575131 |
| 1 | 0.209573 | 0.011596 | 3.090723 | 0.000189 | 0.601642 |
| 1 | 0.085371 | 0.020573 | 0.396168 | 0.000114 | 0.627102 |
| 1 | 0.147607 | 0.012472 | 0.420754 | 0.000135 | 0.639349 |
| 1 | 0.187887 | 0.045957 | 0.008888 | 0.000262 | 0.679573 |
| 1 | 0.073969 | 0.207516 | 0.010205 | 0.000782 | 0.704228 |
| 1 | 0.095339 | 0.030376 | 0.080869 | 0.000589 | 0.707279 |
| 1 | 0.065729 | 0.010705 | 0.314657 | 0.000594 | 0.708929 |

Table 11: Hyperparameter search space for Adam, BERT, batch size 32K.

| Poly Power ($p$) | $1 - \beta_1$ | Weight Decay Rate ($\lambda$) | Learning Rate | Masked LM Accuracy (as fraction) |
|---|---|---|---|---|
| 1 | 0.130375 | 0.000011 | 0.008209 | 0.057367 |
| 1 | 0.130338 | 0.000298 | 0.005776 | 0.057367 |
| 1 | 0.145380 | 0.000295 | 0.005523 | 0.057367 |
| 1 | 0.161747 | 0.031063 | 0.006454 | 0.057367 |
| 1 | 0.211265 | 0.000011 | 0.003328 | 0.057367 |
| 1 | 0.222030 | 0.667330 | 0.000016 | 0.099826 |
| 1 | 0.107412 | 0.000204 | 0.000011 | 0.102016 |
| 1 | 0.248541 | 0.002621 | 0.000027 | 0.102487 |
| 1 | 0.241575 | 0.232839 | 0.000025 | 0.102706 |
| 1 | 0.206147 | 0.234432 | 0.000026 | 0.104275 |
| 1 | 0.190062 | 0.003350 | 0.000059 | 0.115585 |
| 1 | 0.129872 | 0.434452 | 0.000051 | 0.118843 |
| 1 | 0.225584 | 0.000080 | 0.000092 | 0.121976 |
| 1 | 0.218541 | 0.013667 | 0.000113 | 0.126746 |
| 1 | 0.173291 | 0.000074 | 0.000146 | 0.137803 |
| 1 | 0.133697 | 0.000275 | 0.000199 | 0.154843 |
| 1 | 0.204741 | 0.793410 | 0.002174 | 0.279284 |
| 1 | 0.159334 | 0.055297 | 0.000434 | 0.289112 |
| 1 | 0.222569 | 0.060583 | 0.003277 | 0.349611 |
| 1 | 0.102489 | 0.000315 | 0.000507 | 0.350370 |
| 1 | 0.204039 | 0.002175 | 0.000906 | 0.351956 |
| 1 | 0.249884 | 0.017145 | 0.001747 | 0.408409 |
| 1 | 0.166861 | 0.000449 | 0.001317 | 0.414829 |
| 1 | 0.199344 | 0.013357 | 0.001739 | 0.437989 |
| 1 | 0.151030 | 0.000124 | 0.002287 | 0.490207 |

Table 12: Hyperparameter search space for SGD, BERT, batch size 32K.

| Poly Power ($p$) | $1 - \beta_1$ | Weight Decay Rate ($\lambda$) | Learning Rate | Masked LM Accuracy (as fraction) |
|---|---|---|---|---|
| 1 | 0.054284 | 0.000185 | 0.000383 | 0.032528 |
| 1 | 0.062899 | 0.000077 | 0.000248 | 0.032528 |
| 1 | 0.057734 | 0.000225 | 0.000325 | 0.057367 |
| 1 | 0.144769 | 0.000206 | 0.000473 | 0.057367 |
| 1 | 0.093987 | 0.000179 | 0.000401 | 0.057367 |
| 1 | 0.114372 | 0.000224 | 0.000495 | 0.057367 |
| 1 | 0.058382 | 0.000064 | 0.000463 | 0.057367 |
| 1 | 0.149345 | 0.000213 | 0.000294 | 0.061838 |
| 1 | 0.071465 | 0.000186 | 0.000253 | 0.240233 |
| 1 | 0.142337 | 0.000096 | 0.000113 | 0.580597 |
| 1 | 0.134273 | 0.000208 | 0.000216 | 0.629878 |
| 1 | 0.114651 | 0.000124 | 0.000290 | 0.654084 |
| 1 | 0.063694 | 0.000224 | 0.000291 | 0.676234 |

Table 13: Hyperparameter search space for AdamSGD, BERT, batch size 8K.

## B.2 IMAGENET EXPERIMENTS

| Batch size 1024 | | Batch size 8192 | |
|---|---|---|---|
| learning_rate | accuracy | learning_rate | accuracy |
| 0.000100 | 11.908 | 0.000100 | 37.788 |
| 0.000158 | 16.586 | 0.000127 | 42.612 |
| 0.000200 | 19.972 | 0.000394 | 60.916 |
| 0.000316 | 28.038 | 0.001220 | 68.168 |
| 0.000398 | 32.874 | 0.002028 | 69.590 |
| 0.000501 | 37.754 | 0.002972 | 70.116 |
| 0.001000 | 52.020 | 0.002999 | 70.134 |
| 0.001995 | 61.424 | 0.003277 | 70.088 |
| 0.002512 | 63.574 | 0.003442 | 70.488 |
| 0.003981 | 66.834 | 0.004383 | 70.432 |
| 0.005012 | 68.158 | 0.004410 | 70.356 |
| 0.007943 | 70.272 | 0.004417 | 70.314 |
| 0.010000 | 70.760 | 0.006360 | 69.872 |
| 0.012589 | 71.520 | 0.007823 | 69.672 |
| 0.015849 | 71.716 | 0.009804 | 69.538 |
| 0.025119 | 72.554 | 0.010000 | 69.848 |
| 0.031623 | 72.664 | 0.010554 | 69.502 |
| 0.039811 | 72.562 | 0.014409 | 69.068 |
| 0.050119 | 72.656 | 0.021884 | 68.306 |
| 0.063096 | 72.926 | 0.095870 | 25.560 |
| 0.079433 | 72.558 | 0.309611 | 7.162 |
| 0.100000 | 72.658 | 1.000000 | 0.122 |

Table 14: AdaGrad: ImageNet

| Batch size 1024 | | Batch size 8192 | |
|---|---|---|---|
| learning_rate | accuracy | learning_rate | accuracy |
| 0.001000 | 28.636 | 0.005480 | 65.374 |
| 0.001627 | 43.516 | 0.017648 | 73.556 |
| 0.003376 | 59.428 | 0.029705 | 75.218 |
| 0.005493 | 65.994 | 0.048524 | 75.688 |
| 0.008936 | 70.644 | 0.050000 | 75.996 |
| 0.014539 | 73.312 | 0.054569 | 76.108 |
| 0.023654 | 74.638 | 0.059523 | 76.294 |
| 0.038483 | 75.686 | 0.066660 | 76.218 |
| 0.079860 | 76.394 | 0.082167 | 76.278 |
| 0.129926 | 76.612 | 0.108805 | 76.670 |
| 0.165723 | 76.850 | 0.109256 | 76.646 |
| 0.269620 | 76.512 | 0.146364 | 76.442 |
| 0.343903 | 76.232 | 0.182960 | 76.504 |
| 0.438653 | 75.714 | 0.278865 | 76.428 |
| 0.559508 | 75.168 | 0.500000 | 75.356 |

Table 15: SGD: ImageNet

| learning_rate | momentum | weight_decay | objective_value |
|---|---|---|---|
| 0.000010 | 0.886563 | 0.100137 | 0.100 |
| 0.000882 | 0.999000 | 0.088058 | 0.100 |
| 0.007832 | 0.886460 | 0.030099 | 0.100 |
| 0.084027 | 0.867351 | 0.114223 | 0.100 |
| 0.012791 | 0.871190 | 0.762656 | 0.100 |
| 0.001059 | 0.779209 | 0.001784 | 0.100 |
| 0.003035 | 0.951726 | 0.001060 | 0.100 |
| 0.002368 | 0.920451 | 0.006413 | 0.100 |
| 0.043405 | 0.649227 | 0.379924 | 0.100 |
| 0.000064 | 0.987255 | 0.586063 | 0.100 |
| 0.001212 | 0.600000 | 1.000000 | 49.392 |
| 0.028276 | 0.621864 | 0.007048 | 49.550 |
| 0.000010 | 0.600000 | 0.001000 | 50.098 |
| 0.000010 | 0.831437 | 0.103619 | 50.924 |
| 0.000018 | 0.700728 | 0.008109 | 59.668 |
| 0.000058 | 0.999000 | 0.005061 | 66.848 |
| 0.000036 | 0.837616 | 0.001366 | 66.908 |
| 0.001036 | 0.999000 | 1.000000 | 70.878 |
| 0.000144 | 0.889598 | 0.009050 | 72.672 |
| 0.000115 | 0.600000 | 0.036989 | 72.816 |
| 0.000158 | 0.675443 | 0.001000 | 72.840 |
| 0.000628 | 0.606453 | 0.001624 | 73.446 |
| 0.000292 | 0.779432 | 0.008273 | 73.740 |
| 0.000485 | 0.626856 | 0.004095 | 73.744 |
| 0.001786 | 0.877224 | 0.198602 | 73.944 |
| 0.000315 | 0.600000 | 1.000000 | 74.306 |
| 0.000258 | 0.999000 | 0.116977 | 74.562 |
| 0.000572 | 0.600000 | 0.014438 | 74.766 |
| 0.000232 | 0.622821 | 0.192101 | 75.740 |
| 0.000614 | 0.747656 | 0.040759 | 75.968 |
| 0.001000 | 0.799500 | 0.031623 | 76.146 |
| 0.001600 | 0.931992 | 0.058883 | 76.378 |
| 0.001030 | 0.803263 | 0.119977 | 76.398 |
| 0.001189 | 0.848430 | 0.105139 | 76.474 |
| 0.000927 | 0.600000 | 0.101738 | 76.580 |

Table 16: Imagenet:Adam Batch Size 1024

| learning_rate | momentum | weight_decay | objective_value |
|---|---|---|---|
| 0.100000 | 0.638615 | 0.000010 | 0.098 |
| 0.100000 | 0.828515 | 0.000828 | 0.100 |
| 0.100000 | 0.911213 | 0.000494 | 0.102 |
| 0.007128 | 0.714390 | 0.000112 | 3.004 |
| 0.017291 | 0.950635 | 0.000310 | 29.582 |
| 0.000010 | 0.999000 | 0.005794 | 35.810 |
| 0.000010 | 0.999000 | 0.010000 | 36.060 |
| 0.000209 | 0.999000 | 0.000010 | 59.206 |
| 0.000328 | 0.999000 | 0.000010 | 59.268 |
| 0.000010 | 0.690038 | 0.000010 | 62.290 |
| 0.000010 | 0.746399 | 0.010000 | 62.454 |
| 0.000348 | 0.999000 | 0.010000 | 62.476 |
| 0.000013 | 0.797380 | 0.002920 | 64.452 |
| 0.000025 | 0.867802 | 0.000010 | 69.180 |
| 0.000049 | 0.984536 | 0.000378 | 70.680 |
| 0.002565 | 0.991471 | 0.000146 | 70.786 |
| 0.000052 | 0.850277 | 0.010000 | 71.704 |
| 0.000055 | 0.628945 | 0.000010 | 71.826 |
| 0.001567 | 0.697594 | 0.000473 | 71.870 |
| 0.003220 | 0.871350 | 0.000041 | 71.912 |
| 0.001823 | 0.800658 | 0.000302 | 72.114 |
| 0.001373 | 0.601300 | 0.002455 | 72.138 |
| 0.000973 | 0.803842 | 0.000208 | 72.436 |
| 0.000093 | 0.875276 | 0.000058 | 72.690 |
| 0.007858 | 0.964910 | 0.010000 | 72.694 |
| 0.000097 | 0.714558 | 0.000029 | 72.874 |
| 0.000761 | 0.819951 | 0.000481 | 72.914 |
| 0.000979 | 0.802539 | 0.002260 | 72.930 |
| 0.000233 | 0.780565 | 0.000107 | 72.958 |
| 0.000274 | 0.816199 | 0.000225 | 72.978 |
| 0.000197 | 0.773215 | 0.000843 | 72.988 |
| 0.000235 | 0.792918 | 0.000015 | 73.070 |
| 0.000208 | 0.896992 | 0.000010 | 73.150 |
| 0.000427 | 0.796783 | 0.000815 | 73.184 |
| 0.001725 | 0.809302 | 0.004951 | 73.408 |
| 0.000322 | 0.734479 | 0.010000 | 73.538 |
| 0.000647 | 0.851409 | 0.006530 | 73.594 |
| 0.000727 | 0.969539 | 0.010000 | 73.602 |
| 0.000263 | 0.869540 | 0.010000 | 73.658 |
| 0.001207 | 0.844760 | 0.010000 | 73.800 |
| 0.004101 | 0.934473 | 0.010000 | 73.944 |
| 0.002380 | 0.969588 | 0.010000 | 73.984 |
| 0.000771 | 0.856323 | 0.010000 | 74.030 |
| 0.001655 | 0.966147 | 0.010000 | 74.042 |
| 0.002968 | 0.961813 | 0.010000 | 74.196 |
| 0.002168 | 0.952292 | 0.010000 | 74.244 |
| 0.002870 | 0.960954 | 0.010000 | 74.298 |
| 0.000961 | 0.916564 | 0.207196 | 74.380 |
| 0.002261 | 0.869219 | 0.062905 | 74.862 |
| 0.000374 | 0.911343 | 0.376673 | 75.268 |
| 0.000100 | 0.870343 | 1.000000 | 75.430 |
| 0.001082 | 0.944840 | 0.105851 | 75.692 |
| 0.001048 | 0.902476 | 0.093750 | 75.788 |
| 0.000411 | 0.950000 | 0.081495 | 75.856 |
| 0.001008 | 0.881673 | 0.100641 | 75.926 |
| 0.000967 | 0.885877 | 0.097476 | 75.964 |
| 0.000803 | 0.850000 | 0.073826 | 76.048 |
| 0.000905 | 0.850000 | 0.092752 | 76.066 |
| 0.001000 | 0.900000 | 0.100000 | 76.068 |
| 0.000921 | 0.850000 | 0.098263 | 76.096 |
| 0.000913 | 0.850000 | 0.106365 | 76.276 |

Table 17: Imagenet:Adam Batch Size 8192

| Batch size 1024 | | Batch size 8192 | |
|---|---|---|---|
| learning_rate | accuracy | learning_rate | accuracy |
| 0.003162 | 14.846 | 0.001283 | 6.694 |
| 0.003981 | 18.744 | 0.004114 | 17.484 |
| 0.006310 | 31.468 | 0.013190 | 43.236 |
| 0.007943 | 37.240 | 0.068529 | 63.408 |
| 0.010000 | 44.284 | 0.075832 | 64.202 |
| 0.015849 | 54.534 | 0.100000 | 65.428 |
| 0.019953 | 58.262 | 0.103259 | 65.768 |
| 0.025119 | 60.640 | 0.160894 | 67.610 |
| 0.031623 | 63.074 | 0.185812 | 68.104 |
| 0.039811 | 64.830 | 0.200426 | 68.268 |
| 0.063096 | 67.668 | 0.204970 | 68.280 |
| 0.079433 | 68.638 | 0.299255 | 69.638 |
| 0.100000 | 69.144 | 0.303184 | 69.588 |
| 0.125893 | 69.928 | 0.414487 | 70.664 |
| 0.158489 | 70.544 | 0.438243 | 70.708 |
| 0.199526 | 71.102 | 0.524030 | 71.312 |
| 0.251189 | 71.664 | 0.526898 | 71.426 |
| 0.398107 | 72.452 | 0.596139 | 71.438 |
| 0.501187 | 72.864 | 0.626390 | 71.736 |
| 0.630957 | 72.874 | 0.706443 | 72.086 |
| 0.794328 | 73.456 | 0.721966 | 72.044 |
| 1.000000 | 73.502 | 0.726457 | 72.214 |
| 0.876767 | 73.664 | 0.727370 | 72.160 |
| 0.960900 | 73.728 | 0.756109 | 72.164 |
| 1.053106 | 73.742 | 0.760034 | 72.388 |
| 1.264911 | 73.832 | 0.775082 | 72.284 |
| 1.386290 | 73.956 | 0.783008 | 72.274 |
| 1.519316 | 74.256 | 0.796600 | 72.150 |
| 1.824887 | 74.396 | 1.568864 | 0.100 |
| 2.000000 | 74.434 | 10.000000 | 0.100 |

Table 18: Graft SGD#Adagrad (Global): ImageNet

## B.3 CIFAR-10 EXPERIMENTS

| Batch size 128 | | Batch size 512 | | Batch size 2048 | |
|---|---|---|---|---|---|
| Learning Rate | Test Accuracy | Learning Rate | Test Accuracy | Learning Rate | Test Accuracy |
| 0.00075 | 85.418 | 0.00075 | 84.778 | 0.00075 | 82.032 |
| 0.0015 | 88.394 | 0.0015 | 86.778 | 0.0015 | 85.710 |
| 0.003 | 89.814 | 0.003 | 88.254 | 0.003 | 84.502 |
| 0.006 | 88.616 | 0.006 | 87.676 | 0.006 | 82.138 |
| 0.0125 | 87.270 | 0.0125 | 84.070 | 0.0125 | 73.304 |
| 0.025 | 84.822 | 0.025 | 77.164 | | |

Table 19: AdaGrad - CIFAR-10

| Batch size 128 | | Batch size 512 | | Batch size 2048 | |
|---|---|---|---|---|---|
| Learning Rate | Test Accuracy | Learning Rate | Test Accuracy | Learning Rate | Test Accuracy |
| 0.006 | 94.206 | 0.006 | 92.032 | 0.006 | 88.734 |
| 0.0125 | 94.564 | 0.0125 | 92.998 | 0.0125 | 90.724 |
| 0.025 | 94.872 | 0.025 | 93.786 | 0.025 | 91.974 |
| 0.05 | 94.932 | 0.05 | 94.386 | 0.05 | 92.820 |
| 0.075 | 94.784 | 0.075 | 93.638 | 0.075 | 93.372 |
| 0.1 | 94.772 | 0.1 | 93.412 | 0.1 | 88.818 |
| 0.15 | 10.000 | 0.15 | 10.000 | | |

Table 20: SGD (with momentum 0.9 and weight decay 0.0005) - CIFAR-10

| Batch size 128 | | Batch size 512 | | Batch size 2048 | |
|---|---|---|---|---|---|
| Learning Rate | Test Accuracy | Learning Rate | Test Accuracy | Learning Rate | Test Accuracy |
| 0.0125 | 91.610 | 0.0125 | 86.998 | 0.025 | 85.332 |
| 0.025 | 92.878 | 0.025 | 89.764 | 0.05 | 88.136 |
| 0.05 | 93.306 | 0.05 | 91.268 | 0.075 | 89.074 |
| 0.075 | 93.296 | 0.075 | 92.248 | 0.1 | 89.818 |
| 0.1 | 93.420 | 0.1 | 92.360 | 0.2 | 84.770 |
| 0.2 | 10.000 | 0.2 | 83.226 | 0.4 | 46.332 |
| 0.4 | 10.000 | 0.4 | 65.310 | | |

Table 21: SGD#AdaGrad - CIFAR-10

