# OpenReview forum: "Learning Rate Grafting: Transferability of Optimizer Tuning"
_ICLR.cc/2022/Conference — ICLR 2022 Submitted_

### Official Review · Reviewer_4k79 · 2021-10-28

**Correctness:** 3
**Technical Novelty And Significance:** 1
**Empirical Novelty And Significance:** 2
**Recommendation:** 3
**Confidence:** 5

**Main Review:**

The Algorithm and the method are quite simple. Magnitudes are taken from one, and directions are taken from
another.
When you take, D and M#D, in the results, M and M#D are comparable in performance. So there is no clear benefit
to using M#D, when M#D is comparable to M. A consequence of using M#D, they claim, is transferring the
schedules?
But do schedules really generalize well?  Generalize to what? Suppose I change the data and architecture,
can I use this new rate scheduler?  This to be is the question they should address thoroughly.

I think what the authors are proposing is this: If the change is in the optimizer only; how should I change the
learning rate schedule? But, there are additional questions.

Q1. What is the impact of this method on generalization error? If yes, with what confidence?
Q2. Does this new schedule improve the order of convergence? If yes, how?
Q3. Can you discuss general properties of the grafting operator?
Q4. What's the intuition behind transferring schedules (either theoretical or empirical)
and usefulness of grafting?
Q5. why should we expect A#B to perform better than B#A when there are two optimizers A and B, say?

**Summary Of The Paper:**

The authors investigate the entanglements between the optimizer and the learning rate schedule and propose
the technique of optimizer grafting, which allows for the transfer of the overall implicit step size schedule
from a tuned optimizer to a new optimizer, preserving empirical performance. This provides a robust
plug-and-play baseline for optimizer comparisons, leading to reductions to the computational cost of
optimizer hyperparameter search. Using grafting, they discover a non-adaptive learning rate correction to
SGD which allows it to train a BERT model to state-of-the-art performance.

**Summary Of The Review:**

The paper, although is interesting, lacks the technical/empirical novelty to merit publication

---

> ### Author Response · Authors · 2021-11-23
> **Thanks for your review**
>
> Thanks for the review.
>
> > But do schedules really generalize well? Generalize to what? Suppose I change the data and architecture, can I use this new rate scheduler?
>
> We make no claims about the generalization of schedules across datasets/architectures. Our work is concerned with the question of “what accounts for the differing behavior of different optimizers on the same data/architecture setup?”
>
> > Q1. What is the impact of this method on generalization error? If yes, with what confidence?
>
> Our empirical findings are that M#D inherits the training error and end-to-end validation performance of M; thus, we observed that the generalization error was similar to that of M.
>
> > Q2. Does this new schedule improve the order of convergence? If yes, how?
>
> Indeed, when M#D improves upon M, the transferred schedule improves convergence in practice. If D is equivalent to M with a suboptimal learning rate schedule, the provable improvement is obvious. In the general case where D and M output different directions, this improvement is unexplained by current theory.
>
> > Q3. Can you discuss general properties of the grafting operator?
>
> Basic general properties of the grafting operator are discussed very explicitly in Section 3.
>
> > Q4. What's the intuition behind transferring schedules (either theoretical or empirical) and usefulness of grafting?
>
> Theoretically: the differences in performance between adaptive optimizers are explained by their implicit step size schedules (at some parameter grouping granularity).
> Empirically: an immediate application of our strongest result is that a memory-efficient correction to SGD is possible for training state-of-the-art Transformers.
>
> > Q5. why should we expect A#B to perform better than B#A when there are two optimizers A and B, say?
>
> Based on our empirical findings, we should expect M#D > D#M when M > D: it’s more important to get the right sequence of step magnitudes.

---

> > ### Comment · Reviewer_4k79 · 2021-11-23
> > **Reply to authors**
> >
> > Thanks. I'll stick to my initial assessment. Empirical findings about optimizers without generalization characteristics is not good enough, even though the principal idea is interesting.

---

### Official Review · Reviewer_B67d · 2021-10-29

**Correctness:** 3
**Technical Novelty And Significance:** 3
**Empirical Novelty And Significance:** 4
**Recommendation:** 8
**Confidence:** 3

**Main Review:**

As a note, I reviewed a prior submission of this paper. The bulk of my review is the same as in the prior submission (with modifications corresponding to the modifications in the paper).

## Originality

The paper is original, performing an experiment that I have not seen in the literature, and satisfactorily discusses background and related work.

## Quality

The experiments performed within the paper are of high quality, satisfactorily demonstrating the main claims of the paper, that groupwise grafting results in performance equivalent to the better-tuned optimizer that the learning rates are derived from.

My main qualm with the experiments is the lack of motivation for the specific experimental configurations. Instead, the results of several disparate experiments in disparate settings are reported, making it hard to reason about the generality of the results. Specifically:
- There are no reported results for global grafting on BERT, other than a statement that it performs "significantly worse". It would be helpful to see these results (in the main body or appendix) to understand how much worse and the tradeoffs between global and layer wise grafting.
- The "simplifying the discovered schedule" section also feels arbitrary. The correction scheme is described as "simple", which it is, but it is also a bespoke correction for a single experiment which is not validated to transfer to other settings. It's not obvious what the contribution of this section is beyond that of the rest of the paper.

I would also appreciate further validation that the differences in directions between M and D are significant on the layerwise and global scale: given that M # D == M when using per-weight groups, it seems possible that when using smaller groups that are e.g. the size of layers, M # D \approx M. It would be worthwhile to show that the group-wise directions of M and D are significantly different. (my expectation is that they are essentially orthogonal, but this is worth validating).

## Clarity

On the whole, the paper is very clear, with enough details to easily understand and reproduce the experiments reported in the paper. However, I do have the following questions and suggestions for improvements:
- An explicit statement of the precise hypothesis being tested would significantly improve the paper. As-is, the results are possible to interpret in many different ways (e.g., that grafting either gives the performance of M, gives the better between M and D, or something else); and explicit statement of the hypothesis along with additional experiments to precisely test the bounds of that hypothesis would result in a much more precise takeaway from the paper.
- Section 2.1: how does this deal with stochasticity in the gradient estimate? Based on Eq. 1, it seems like $g_t$ is the gradient for a minibatch, but the algorithm does not specify minibatch selection.
- The notation of M#D and granularity could be improved; I kept having to scroll up and try to figure out which refers to magnitude and which refers to direction, and the granularity is specified separately in text.
- The plots (specifically, Fig 2,3) are very small and hard to read.

## Significance

Though I am not an expert in optimizers, the results seem like a useful step towards empirically understanding the contrast and resultant difference in real-world performance between different optimizers.

 # Update after author response

Thanks to the authors for the response, especially the explicit statement of the research question. I do understand that the two experiments that I raised slight concerns with ("simplifying the discovered schedule" and global grafting for BERT) are not intended to be reusable methodologies for training networks in the future, but my concern is that without clearly stated methodologies/justifications for these one-off experiments or broader replication, the results may just be the result of chance (having continued to try variants of the approach until one worked) rather than supporting the broader claim that the (effective) learning rate schedule is the primary driver of accuracy.

Regardless, my critiques are ultimately minor, and I do still think that the paper should be accepted. My main point of disagreement from the other reviewers is that I believe this paper is a significant contribution even if nobody ever uses the technique to train a network to deploy to end-users, because the paper's contribution is to the empirical understanding of optimizers rather than to the actual usage of optimizers. Because of that, I think that a slightly lower methodological bar is reasonable -- the paper is presenting a new finding moreso than a new technique, and we should not require that the approach works in all settings or even that the authors describe methodologies for finding similar results in new settings (e.g., with different hyper parameters or choices of M and D).


**Summary Of The Paper:**

The authors propose learning rate grafting as a method to explore the power and dynamics of optimizers. Learning rate grafting partitions the parameters of the networks into groups, and for each group takes the direction of the weight update from one optimizer and the magnitude from another optimizer. The paper then shows that grafting allows for achieving the performance of a tuned optimizer using that tuned optimizer's group-wise magnitudes along with an untuned optimizer's group-wise directions.

**Summary Of The Review:**

Accept. While I am not entirely satisfied with the motivation behind the choices of reported experiments and the missing statement of a falsifiable hypothesis to test, the paper proposes an interesting experiment and provides a technically correct evaluation of the claims in the paper.

---

> ### Author Response · Authors · 2021-11-23
> **Thanks for the review, support, and constructive feedback**
>
> Thanks for the review, support, and constructive feedback, which we have incorporated in the updated manuscript.
>
> > An explicit statement of the precise hypothesis being tested would significantly improve the paper.
>
> We appreciate this feedback, and have revised the exposition. At the end of Section 2, we’ve expanded the paragraph on “entanglements between preconditioners and learning rates” to walk through the motivation behind grafting, and the simplest version of the research question (“can the effectiveness of an adaptive optimizer be explained by its induced learning rate schedule?”). The discussion in Section 3 is now more connected to this research question.
>
> > The "simplifying the discovered schedule" section also feels arbitrary.
>
> In our view, the quantization of schedule corrections serves as a proof-of-concept towards a future empirical goal (finding a concise correction to SGD that works broadly for Transformers), which is outside the scope of this work. This would be a significant and immediately useful standalone contribution, which we would expect to require empirical diagnostics beyond grafting and architecture-specific insights.
>
> > It would be helpful to see these results (in the main body or appendix) to understand how much worse and the tradeoffs between global and layer wise grafting.
>
> To clarify, there is no such tradeoff in the BERT experiments-- we were unable to observe any nontrivial training progress (over SGD) using global grafting, hence the omission. This corroborates the empirical difficulty of getting SGD to train state-of-the-art Transformers-- a global learning rate schedule does not suffice to account for the benefits of adaptive methods.

---

### Official Review · Reviewer_JLsZ · 2021-10-31

**Correctness:** 2
**Technical Novelty And Significance:** 2
**Empirical Novelty And Significance:** 2
**Recommendation:** 3
**Confidence:** 3

**Main Review:**

The paper presents an interesting technique of grafting for the problem of step size hyperparameter tuning, and opens up questions as to the power of simple per-learning rate schedules. The strengths of the paper include performing analysis on state of the art benchmarks (ImageNet, CIFAR-10 for image classification and Wikipedia, Books for BERT pertaining). Another strength is assessing the context of transferring implicit step size schedules to another optimizer, for assessing global and per-layer variants, and for assessing simple learning rate discovery.

However, empirical evidence was not sufficient to be conclusive on the methods presented. For example, the results were on 2 tasks  (image classification with ResNets and BERT pretraining) with 2 datasets each, and 2 batch sizes (8192 and 32768). Further, (for example) , Figure 2 displays results for the “best trial” performance, but does not include error bars. Adding error bars and the empirical computational cost reductions would be useful in better supporting the claims.

On several steps, potentially concerning results were identified but not adequately justified, and without more sufficient empirical results, the reader is uncertain if these are indications of more serious flaws in the approach. (e.g., for BERT (p.6), the global version of grafting were signficantly worse and “have been omitted” and (p. 14) despite these exponentially large correction ratios, the grated optimizer converges, but this “curious phenomenon” was left to future work).

The study also does not present theoretical underpinnings for the technique that would be helpful for understanding if indeed the results could be more widely applicable.

The study presents the approach M#D, but does not give guidance on how M or D might be selected more generally for other tasks, or how one might assess tasks to decide on M and D. There are also decisions for parameters that are not explained and it is not clear now sensitive results may be to these decisions. For example for learning rate discovery (p.8),  authors describe choosing the layer-wise multiplier as the median of individual corrections for the first 2000 iterations, or discretizing to the nearest power of ten. Providing additional guidance and/or theoretical underpinnings for such choices would be useful.

**Summary Of The Paper:**

The authors report on a technique to address learning rate hyperparameter tuning for deep learning referred to as optimizer grafting. Specifically, the paper proposes a meta-algorithm (referred to as M#D) that blends the steps of two optimizers by combining the step magnitude of one (M) with the direction of the other (D). The technique of optimizer grafting allows for the transfer of the overall implicit step size schedule to a new optimizer, resulting in reductions in computational cost of optimizer hyper parameter search. The second primary result is leveraging the technique to identify a non-adaptive per-layer learning rate correction to SGD which allows it to train a BERT model to state-of-the-art performance. Analogous results are presented for vision models for global (non-per-layer) schedules for AdaGrad.

The authors describe grafting meta algorithm  (M#D) as, at each iteration, M#D feeds the same input (w_t, g_t) to both M and D which manage their states independently and produce w_M, w_D. Then the norms of the steps each would have taken is computed, and used to combine M’s magnitude update with D’s direction update. Partitioning is managed to implement global versus per-layer grafting.2 optimizer hyperparameter searches with the same computational budget, but different performances.

The authors present an empirical study on the transfer of implicit step size schedules between optimizers, comparing SGD and Adam to Adam#SGD for task of BERT pre-training. They show that Adam#SGD is able to achieve performance at/near Adam.

The paper also presents results for image classification for ImageNet and CIFAR-10), for AdaGrad, SGD, and SGD#AdaGrad, showing  SGD#AdaGrad outperforms SGD and AdaGrad. However, without error bars, it is hard assess the actual results.

Finally, the paper shows results for grafting distilling a non-adaptive correction to D, eliminating the need to run M in parallel - that is, transferring a global, time-dependent non-adaptive multipliers for the learning rate. The results show for the global variant for ResNET (SGD, AdaGrad), the discovered learning rate is comparable to the one used on SGD and achieves a top1 accuracy of 72.46. For the per-layer variant, learning rate schedule enables a simple per-layer step size correction without adaptive preconditioning. The authors present proof-of-concept results for simplifying the discovered schedule as a way to support the robustness of their transfer approach.


**Summary Of The Review:**

Overall, the paper presents an interesting technique and opens interesting questions, but without further empirical results or theoretical underpinnings, the results presented are insufficient to assess how generalizable the findings might be.

---

> ### Author Response · Authors · 2021-11-23
> **Thanks for your review.**
>
> > the global version of grafting were significantly worse and “have been omitted”
>
> We are not sure why this is a concerning omission-- this is a negative result (the least granular version of grafting fails to transfer performance from M to D), and we found the evaluation metrics/training curves to be uninformative.
>
> > despite these exponentially large correction ratios, the grated optimizer converges
>
> Given that the grafted optimizer does converge, we are also not sure why this is of concern. Evidently, the Adam preconditioner has very large multipliers for certain parameters, but the learning rate transfer experiments show that it doesn’t matter whether or not they’re included.
>
> > does not give guidance on how M or D might be selected more generally for other tasks
>
> This question of how to select an optimizer for an architecture/task is far more general than what we tackle in this paper, and far from resolved. Our main results can be interpreted as providing one new piece of insight towards this mystery: “the choice of learning rate schedule (at some granularity) can at least partially explain discrepancies attributed to optimizer choice”.
>
> > There are also decisions for parameters that are not explained and it is not clear now sensitive results may be to these decisions. For example for learning rate discovery (p.8), authors describe choosing the layer-wise multiplier as the median of individual corrections for the first 2000 iterations, or discretizing to the nearest power of ten.
>
> These decisions are necessarily heuristic, since we are discovering learning rate corrections based on the nonstationary time series associated with model training.
>
> Firstly, as we have mentioned in the paper, we are not proposing this scheme (medians or rounding) as a general recipe to transfer schedule, but rather as a means to show that there exists a relatively coarse non-adaptive schedule which is sufficient to have SGD train as well as Adam. Another way to interpret the result is that it suggests that SGD + a few corrections can train as well as Adam (which uses double the memory of SGD) -- reducing the actual memory requirement as well as conceptually reducing the search space for further algorithm design. Please also refer to the response to other reviewers who have raised similar points.
>
> Further, we note that the effectiveness of the power-of-10 discretizations suggests that the learning rate discovery technique is robust to rounding, strongly suggesting that varying the “median of first N iterations” parameter is not a source of sensitivity.

---

### Official Review · Reviewer_9GnN · 2021-11-02

**Correctness:** 2
**Technical Novelty And Significance:** 2
**Empirical Novelty And Significance:** 2
**Recommendation:** 3
**Confidence:** 5

**Main Review:**

S1. A large amount of experiment is conducted and plenty of result is shown in appendix.

S2. A novel optimizing mode of grafting two different optimizers is proposed.

W1. The paper structure is strange. I recommend to read some published proceedings to try to make this paper more clearly.

W2. Some format maybe not legal. Such as the caption of table and content in Page 16.

W3. The theory is not reasonable. In other word, you just tell me you do it like this but not why it’s reasonable. Actually, I don’t think ADAM#SGD will be better than ADAM. ADAM calculates the update direction according to loss function. In a multi-dimensional space, this direction is composed of the value of each gradient and positive or negative(symbol) of each gradient. However, you change the symbol of some parameters’ gradient according to SGD. This is why? I’m confused. In my view, this method is more like a SGD with multiplying a large const to its gradient.

W4. I have a question, how to compute the norms (||w_m-w_t ||)/(||w_D-w_t ||). Is ||w_m-w_t || calculated with all the parameters in neural network? If not, I think Figure~1 is a wrong example, cause M#D will step to different direction with D in multi-dimensional space.

W5. The results shown in tables are not strong enough.

**Summary Of The Paper:**

This paper proposes a method called optimizer grafting. It uses two optimizers in one training session. One is to decide the update direction of parameters, and the other is to decide the update stride of parameters. This paper proposes a new optimizing mode and take a large amount of experiment exploration.


**Summary Of The Review:**

Even though some idea is interesting, the theoretical work in this paper is insufficinet.

---

> ### Author Response · Authors · 2021-11-23
> **Thanks for the review - We request the reviewer to read the paper carefully**
>
> Thanks for the review.
>
> > W1. The paper structure is strange. I recommend to read some published proceedings to try to make this paper more clearly.
>
> Could the reviewer please clarify what they found ‘strange’ about the paper structure?
>
> > W2. Some format maybe not legal. Such as the caption of table and content in Page 16.
>
> Page 16 is in the appendix - as far as we know there are no format restrictions there.
>
> > W3. The theory is not reasonable.
>
> Though we hope that the empirical findings will stimulate future theoretical inquiry, we have not attempted to provide theory in this paper. There remains a salient gap between well-understood theoretical abstractions of optimization and state-of-the-art neural network training dynamics; this investigation is purely an empirical study of the latter.
>
> > I don’t think ADAM#SGD will be better than ADAM.
>
> We are confused about the basis of this belief-- our main empirical finding is that M#D inherits the empirical performance of M. Please note that we are not making a claim about the superiority of a grafted optimizer anywhere in the paper. This paper is not about proposing a new optimizer or an optimizer that’s better than others.
>
> > This direction is composed of the value of each gradient and positive or negative(symbol) of each gradient. However, you change the symbol of some parameters’ gradient according to SGD”
>
> Indeed Adam proposes a different direction than SGD. This to the best of our knowledge has nothing to do directly with positive or negative ‘symbols’ (interpreting symbols as signs) – in fact as such in the case when momentum is 0 the per component signs of both Adam and SGD will remain the same. However, this point is moot and irrelevant – the point of the paper is that Adam#SGD is much better than SGD (assuming we are talking about the BERT experiment) – and our experiment shows that the intrinsic learning rate schedule of Adam can be transferred both adaptively and non-adaptively over SGD making it competitive with Adam.
>
> > “how to compute the norms?”
>
> We specify clearly that there are two modes of norm computation that we do – global, i.e. an ell_2 norm over all the parameters and layer-wise, an ell_2 norm over parameters only in a layer. Figure 1 is specifically drawn for the global case and that is clear. For the layer-wise the figure is a schematic for a particular layer. In our opinion all this is completely clear and the reviewer needs to read the paper more carefully.

---

### Decision · Program_Chairs · 2022-01-20

**Decision:**

Reject

**Comment:**

The paper proposed Trained ML oracles to find the decent direction and step size in optimization. The process they call grafting. Reviewers raised several concerns about the reliability of ML oracles in general settings which is valid. The rebuttal could not convince the reviewers to change their opinion.  Ideally for an empirical only paper with heavy reliability on ML for critical decisions, to meet the high bar of ICLR there must be several experiments (5-10 datasets or more) on diverse datasets and settings. Also, there should be discussions on when and how the method fails and related discussions. In that sense the paper does not meet the bar for publication.